# Drivers of rodent community structure in an Urban National Park, Kenya

**Immaculate M. Mungai**[1,2], **Nathan Gichuki**[1], **Dorcus A.O. Sigana**[1], **Benard Agwanda**[2], **Patrick Chiyo**[3], **Vincent Obanda**[4], **Olivia Wesula Lwande**[5,6]*

1 Department of Biology, University of Nairobi, Nairobi, Kenya, 2 Department of Mammalogy, National Museums of Kenya, Nairobi, Kenya, 3 Department of Biology, Duke University, Durham, North Carolina, United States of America, 4 Department of Veterinary Science and Laboratories, Wildlife Research and Training Institute, Naivasha, Kenya, 5 Department of Clinical Microbiology, Umeå University, Umeå, Sweden, 6 Umeå Centre for Microbiology Research, Umeå University, Umeå, Sweden

* olivia.lwande@umu.se

## Abstract

Nairobi National Park (NNP) is among Kenya's most vulnerable ecosystems, experiencing significant pressure from urbanization. Rodents, which are sensitive to environmental changes, are considered bioindicators of ecosystem health, and their population dynamics can be used to assess ecosystem pressures such as urbanization. This study assessed the rodent community structure in NNP to understand the effects of various urban pressures by examining the relationships between rodent diversity, richness, and abundance with vegetation types and metrics, seasonality, and habitat disturbances. The capture-mark-release method was used to trap rodents from 15 sites in Nairobi National Park's savannah, forest, and riverine vegetation types during the dry and wet seasons. The diversity, species richness and abundance were determined from the trappings. From 56 rodents trapped, five species were identified namely: *Lemniscomys striatus, Hylomyscus sp, Rattus rattus, Mus mus* and *Otomys tropicalis.* Rodent diversity at NNP was low (Simpson=0.7130; Shannon Weiner=1.40; Brillouin index=1.27) while Pielou's species evenness, was moderate=0.44 indicating near equity in species distribution. Univariate Generalised linear models showed that rodent abundance was influenced by season, vegetation type, and vegetation metrics. The multivariate model indicated that rodents were more abundant in the wet season compared to the dry season, and that abundance was also positively associated with increased tree and shrub densities. Rodent species richness was positively associated with higher tree density, while vegetation types influenced rodent species diversity. Rodent abundance was influenced by vegetation type, vegetation metrics (density and cover), and season. Human disturbance had no effect in both models. It was observed that the diverse anthropogenic activities occurring in NNP, do not significantly influence rodent abundance compared to the measured biotic and abiotic factors. This first rodent survey in this Park provides preliminary data for continued monitoring of this ecosystem.

**Data availability statement:** All relevant data are within the manuscript and its Supporting information files.

**Funding:** Initial of author who received the award> OWL The study was supported by the Swedish Research Council, Research network grant with a focus on Swedish research links, Registration number 2021-05307. URL: https://www.vr.se/english.html The funders had no role in study design, data collection and analysis, decision to publish, or preparation of the manuscript.

**Competing interests:** The authors have declared that no competing interests exist.

## Introduction

Although, urban protected areas, green public spaces adjacent to large human centers aimed at legally conserving, preserving and protecting biodiversity, provide for recreation, a refuge for wildlife, preservation of cultural values, and provision of ecosystem services [1–4], their existence is wrought with external pressures linked to infrastructural developments in pursuit of social and economic objectives [3]. Human-focused developments and associated pressures can alter resource availability, impact wildlife and plant community health, and in turn influence ecological integrity, the composition, structure, and functioning of an ecosystem [5]. Ongoing ecological monitoring of such ecosystems is thus crucial for their sustainability.

The use of bioindicator species for monitoring ecological changes is underpinned by the assumption that they interact with other facets of the ecosystem and thus provide an indicator of its health [6]. The choice of suitable bioindicator species is often dictated by the goals of the monitoring exercise [7]. Communities of small mammals have been used as bioindicators in varying environmental contexts as they are easy to identify, live in small patches with survival correlating to changes in vegetation, reproduce quickly, and are easy to sample [8]. Besides being the most diverse mammalian taxon [9], rodents are integral components of diverse ecosystems (e.g., forests, savannah, riverine), and play critical roles in the ecosystem functioning, ranging from bottom-up (e.g., seed dispersal; [10]) and top-down (e.g., crucial prey [11]) processes. However, rodents are sensitive to ecosystem changes (both naturally and anthropogenically caused), which affect their abundance, distribution, diversity, and richness [12,13], making them suitable bioindicators [10,11].

The Nairobi National Park (NNP), is a unique urban protected area in the city of Nairobi, that has recently been impacted by several infrastructural developments, including the construction of a tarmac road highway along the southern bypass, the Standard Gauge Railway line and uncontrolled human settlements within and outside its park boundaries resulting in habitat disturbances incompatible with its founding objectives [3]. The continued encroachment of NNP [7], alongside other threats, including poaching, human-wildlife conflicts, and loss of dispersal areas have had deleterious effects on wildlife [5–8] and plant populations [9]. Several studies have been undertaken to monitor different facets of NNP including insects [14], wildlife population trends [15,16], impact of land-use changes [17–19], wetlands [20], and vegetation [21]. These have however been temporally and spatially incongruous to reveal the underpinnings of current changes associated with ecological disturbance and yet its continued vulnerability requires systematic ecological monitoring for its sustainability.

How the habitat modifications (e.g., creating edges) caused by infrastructural developments, coupled with other anthropogenic pressures towards NNP have impacted the rodent community structure is unclear because of inadequate data. In contrast to other Kenyan protected areas where rodent surveys are routine [1–5], only one survey was conducted in NNP, 56 years ago in which *Crocidura fumosa* and *Mastomys coucha* were the only rodents captured in 4,320 trap nights [6]. This study seeks to understand the association between habitat structures and rodent community structures, whereby we hypothesize that habitat variability influences rodent community structure (e.g., rodents will be abundant in 'edge' habitats such as near human settlements or Park boundary). Therefore, the overall objective of this study was to characterize rodent species diversity and abundance in NNP and determine the influence of seasonality, vegetation metrics, and habitat disturbance on the rodent community structure.

## Materials and methods

### Study area

The present study was conducted in Nairobi National Park (NNP), located seven kilometers from the city of Nairobi (Fig 1). The park has a size of 117 km². NNP is contiguous with the

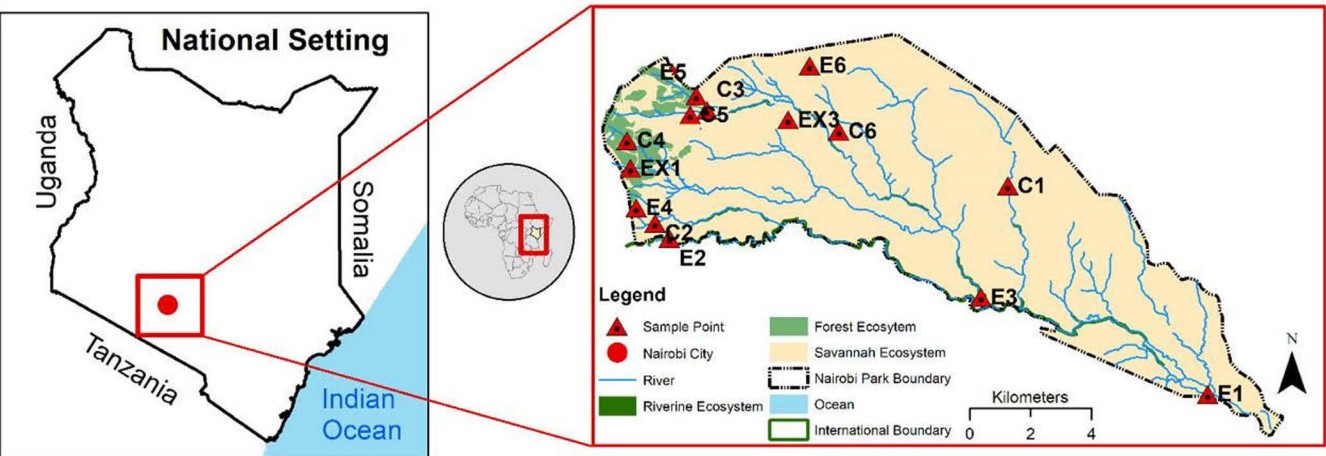

**Fig 1. The left map is for Kenya, which is inside the map of Africa (middle).** The blown-out map (right) is for Nairobi National Park, where the study was conducted. The sampling sites in Nairobi National Park include sites at the Edges (E1, E2, E3, E4, E5 & E6), representing control for the edge habitats (C1, C2, C3, C4, C5 & C6) and additional non-categorized sampling sites within the park (EX1, EX2 & EX3). The names of the sites are as follows; E1 Athi Basin, E2 Bangla, E3 Mokoiyet, E4 David Sheldrick, E5 Clubhouse, E6 Asian Settlement; C1- Lion Dip, C2- Kingfisher, C3- Nangolomon Dam, C4- Nairobi Tented Camp, C5- Nangolomon Circuit, C6- Southern Bypass; EX1- Nairobi Tented Camp, EX2- Hyena Dam and EX3- Park Point 2B).

Athi-Kapiti ecosystem on its southern border, which serves as a dispersal area for wildlife despite mushrooming satellite towns, infrastructural development, and fencing. Wildlife, mainly zebra and antelopes, migrate out of NNP to the Athi-Kapiti ecosystem during the dry season between August and October [15]. The annual amount of rainfall varies between years, but averages 808 mm, with a range of 366–1697 mm, having a peak from March to May and a decline from October to December [15]. It was reported that between 2000 and 2017, the intra-seasonal rainfall variability increased [22] and temperature fluctuates between the warm season in December to April, and the cool period extends from June to August [23]. Relative humidity oscillates between 55% during the day and increases to 80% at night [23]. The park is characterized by open grasslands dominated by *Pennisetum mezianum, Setaria phleoides, Themeda triandra, Digitaria macroblephora* grass species, and scattered low-canopy *Vachellia drepanolobium* [22]; open woodland forest, mostly on the elevated areas on the west as well as riverine woodland along the streams in the park. Large mammals in the park include herbivores (antelopes, buffalo, giraffes, wildebeest, zebra, rhinoceros), and carnivores (hyenas, lions, jackals).

## Rodent community survey

Rodent sampling was performed from December 2020 to June 2021. Rodents were sampled using a stratified random design. This design employed the line transect approach, which provides a better resolution of community structure for a given effort to sample rodents in the three distinct vegetation types in NNP (Savanna open grassland, dry upland forest, and riverine forest or woodland). In addition, the line transect is sensitive to pick diversity, relative abundance and species richness of small mammals compared to a grid system approach [24,25]. We sampled rodents in NNP by placing traps randomly along the park boundaries, areas close to human settlements inside the park, as well as sites away from the park boundary and human settlements. Park boundary and human settlements represented 'edge' habitats. To determine the effect of 'edge habitat on rodent community structure', each sampling transect at the 'edge' had a control sampling point with traps placed away from 'edge habitats'. 20

Sherman traps were placed at each sampling station throughout the 100m line transect. The trap night effort was based on the number of traps used multiplied by the number of trapping nights. Peanut butter and whole-grain oats were used as baits in the traps, which were deployed for three consecutive nights before relocating traps to other sites. The Capture-Mark-Release method was employed, whereby upon trap inspection each morning, the trapped rodents were removed from the trap, identified, marked (ear snip) and released. The identification features of importance were mass, sex, reproductive condition and species' specific traits such as hair/fur and pelage pattern, tail structure and length, body size. The species were also recorded at this stage too.

## Vegetation variables

We sampled vegetation in NNP to estimate the densities of trees and shrubs, the ground cover, and the herb layer. We estimated the tree density using the point-centered-quarter (PCQ) method [26], whereby trees (plants >5 m in height) and shrubs within quadrats placed at equidistance over the rodent-sampling transects were enumerated and identified to the lowest taxa. We estimated shrub density (plants <5 m in height) within quadrats placed at equidistance along rodent-sampling transects by counting the shrubs and identifying them to the lowest possible taxa. We determined the ground cover by visual assessment of the percentage of ground covered by vegetation within a 30 x 30 cm wire quadrat, thrown over the rodent trapping transects in the three vegetation types in NNP (savannah, riverine, and forest).

To collect herbaceous data, we established a 1 m x 1 m plots along the transects in each vegetation type (savannah, riverine and forest) and estimated the percentage cover of herbs.

## Sampling sites

We sampled 15 sites according to the three vegetation types in NNP (Fig 1) namely, the Asian Settlement, Bangla, Ex1, Kingfisher and Southern Bypass, Club House, David Sheldrick, Ex2, Nairobi Tented Camp, Nangolomon Circuit), Athi Basin, Hyena Dam, Lion Dip, Mokoiyet, and Nangolomon Dam. These sites were further classified as human-disturbed areas, control (undisturbed areas in the park with similar vegetation characteristics and at an average distance of 3 km from the disturbed site), and the Park (general undisturbed area). The edge or disturbed sites consisted of (Asian Settlement, Athi Basin, Bangla, Club House, David Sheldrick, Mokoiyet, and Southern road bypass), while the control sites consisted of (Kingfisher, Lion Dip, Nairobi Tented Camp, Nangolomon Circuit, Nangolomon Dam). In the areas classified as Park (undisturbed), additional site named Ex1, Ex2 and Hyena Dam was sampled for each vegetation type; savanna, forest and riverine.

## Data analysis

**Rodent species diversity, richness, evenness and abundance.** To describe the community structure of the rodents in NNP, we used three diversity indices - the Simpson diversity index, Shannon Weiner diversity index, and the Brillouin diversity index. The Shannon-Wiener Index (H') assumes that individuals are randomly sampled from an infinite population and that all taxa are represented in the sample. It is defined using the following equation:

$$H' = \sum \left( \frac{ni}{N} \times ln\frac{ni}{N} \right)$$

Where - ni is the number of individuals of each of the i species, and N is the total number of individuals at each site.

Values of H′ can range from 0 to 5, although they typically range from 1.5 to 3.5.

The Brillouin Index (HB) is a modification of the Shannon-Wiener Index that is preferred when samples are likely not to have been sampled randomly:

$$H_B = \frac{lnN! - \sum \ln ni!}{N}$$

The Simpson's Index ($\lambda$), which is the probability that two individuals drawn at random from an infinitely large community will be different species, is a measure of dominance (which is based on a threshold of percent species cover per site) and as such weighs towards the abundance of the most common taxa. Simpson's Index is expressed as the reciprocal (DS=1−$\lambda$), hence a measure of diversity, where higher values represent higher diversity. It is less sensitive to rare species than the Shannon-Wiener Index. Simpson's index ranges from 0 to 1, and is defined by the following equations:

$$\lambda = \sum \frac{ni(ni-1)}{N(N-1)}$$

$$D^s = 1 - \sum \frac{ni(ni-1)}{N(N(-1)}$$

These indices were calculated using the "vegan" package of the R statistical software version 2.5–6 [27]. In addition, the differences in diversity across vegetation types, disturbance types and seasons were tested using the Hutcheson student t-tests from the "ecolTest" package [28] in the R software for statistical computing.

## Density of trees and shrubs and herbaceous percent cover

We determined each location's density of trees and shrubs, the percent herbaceous cover, and the dominant plant species ($\geq$ 80% cover), first, to characterize their relationships with vegetation type and human habitat disturbance. Second, we tested the direct influence of variation in tree and shrub densities and percent herbaceous cover on rodent diversity.

We used Principal Components Analysis (PCA) to classify the vegetation types or habitat disturbance by plotting scores in a two-dimensional biplot to characterize habitat variation [29] based on tree and shrub density, herbaceous cover, and dominant species in relation to vegetation type and human disturbance. We used PCA to distinguish habitat classes and important traits because it is a scaling method that decreases the dimensions of complex multivariate data, such as shrub and tree densities, herbaceous cover, and dominating plant species [30,31].

Additionally, the differences in mean densities of trees and shrubs between vegetation type and human disturbance were tested using Analyses of Variance (one-way ANOVA). However, the differences in herbaceous cover among vegetation types, disturbed and control sites were tested using Kruskal-Wallis tests. The vegan package was used in PCA while the R software [32] was used for ANOVA and Kruskal-Wallis tests.

We used a Generalized Linear Model framework with a Poisson family and a log-link function [33,34] to test the influence of habitat metrics. Metrics including, tree and shrub densities, herbaceous cover, vegetation type and habitat disturbance on the rodent species richness, abundance, and diversity (Shannon Weiner diversity index) per site, were incorporated as dependent covariates in three independent models.

The following independent covariates (predictor variables) were used; vegetation metrics (tree density, shrub density, herbaceous vegetation cover), vegetation type (forest, savannah, and riverine vegetation), disturbance (disturbed, control and park), and season (wet and dry periods). Numeric variables, particularly vegetation metrics, were standardized to stabilize the variance of coefficients and to provide unbiased hypothesis testing.

Univariate analysis followed by multivariate analyses and model selection was performed. Model selection was attained by computing the coefficients of all possible simple and complex model combinations using the 'MuMIn' package in the R statistical software (Barton 2020). The best model was selected based on Akaike Information Criteria (AIC); the most parsimonious or best model being one of the smallest AIC value [35].

### Ethical statement

The study was approved by Kenya Wildlife Service, permit number KWS/BRP/5001. The study was non-destructive because non-lethal traps were used for capture and release method.

## Results

### Rodent species diversity (richness, evenness) and abundance

The total trap nights were 2700 while 65 trap nights were ineffective because of traps that triggered themselves and failed to capture a rodent or traps that captured non-target species like birds, leaving 2635 effective trap nights. A total of 56 individual rodents were captured consisting of five species dominated by *Lemniscomys sp* (43%), followed by *Rattus rattus* (23%) and *Otomys tropicalis* (20%) while *Mus mus* (5%) and *Hylomiscus sp* (9%) were less common. The overall rodent diversity in NNP was low (Simpson = 0.7130102; Shannon Weiner = 1.40, Brillouin index = 1.27). Pielou's species evenness, J, was moderate (0.44), indicating nearly average equity in species distribution.

Despite identical trapping efforts, rodent abundance and species richness varied across the 15 sampling sites (Fig 1) investigated in this study. No rodent was captured from the Southern Bypass, Nangolomon Circuit, Lion dip, and Asian settlement sites. However, the highest number of rodents was captured in the permanent housing areas (David Sheldrick and the Nairobi-tented campsite). Higher species richness per site of rodents were caught at permanent housing areas in the park (Club House, David Sheldrick, and Nairobi Tented Camp sites) compared to undisturbed areas of the park (Ex1, Ex2, Hyena dam, Kingfisher, Athi basin, Bangla, and Nangolomon dam) where a single species was caught. The site with the highest rodent diversity was Nairobi Tented Camp (DS = 0.57, H'= 0.94 and HB = 0.69) followed by Club House (DS = 0.56, H' = 0.95 and HB = 0.56). The David Sheldrick site had a low species diversity (DS = 0.30, H'= 0.48 and HB = 0.40). The rest of the park had zero diversity.

Forests had a higher species diversity, richness and abundance than either savannah or riverine vegetation, however, species evenness was similar between forest and savannah vegetation (Fig 2). There was a statistically significant difference in Shannon Weiner diversity between forest vegetation and savannah (forest = 1.50, savannah = 0.54; Hutcheson t-statistic = 5.98, df = 21.27, P <0.0001) or riverine vegetation (forest = 1.50, riverine = 0.377; Hutcheson t-statistic = 4.68, df = 9.91, P = 0.0009). There was, however, no difference in diversity between savannah and riverine vegetation (savannah = 0.540 and riverine = 0.377; Hutcheson t-statistic = 0.611, df = 14.03, P = 0.551).

Human disturbed sites or habitat edges had a slightly higher Shannon Weiner rodent diversity index than the control sites (Fig 3), but the difference was not statistically significant (edge = 1.145, control = 1.022; Hutcheson t-statistic = 0.91, df = 50.38, P = 0.370).

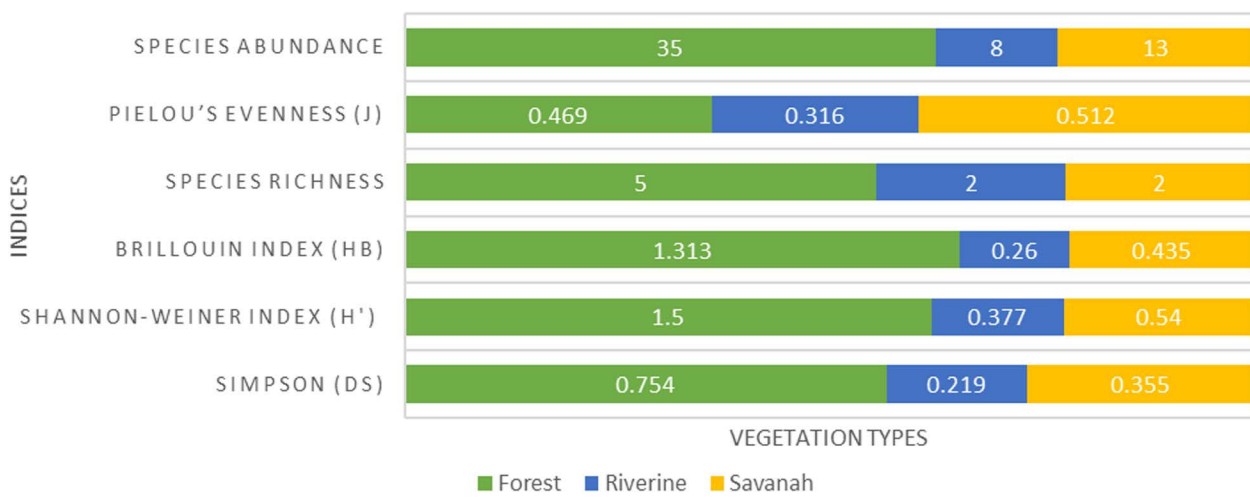

**Fig 2. The diversity, species richness, evenness and abundance of rodents in the forest, savannah and riverine zones of Nairobi National Park.**

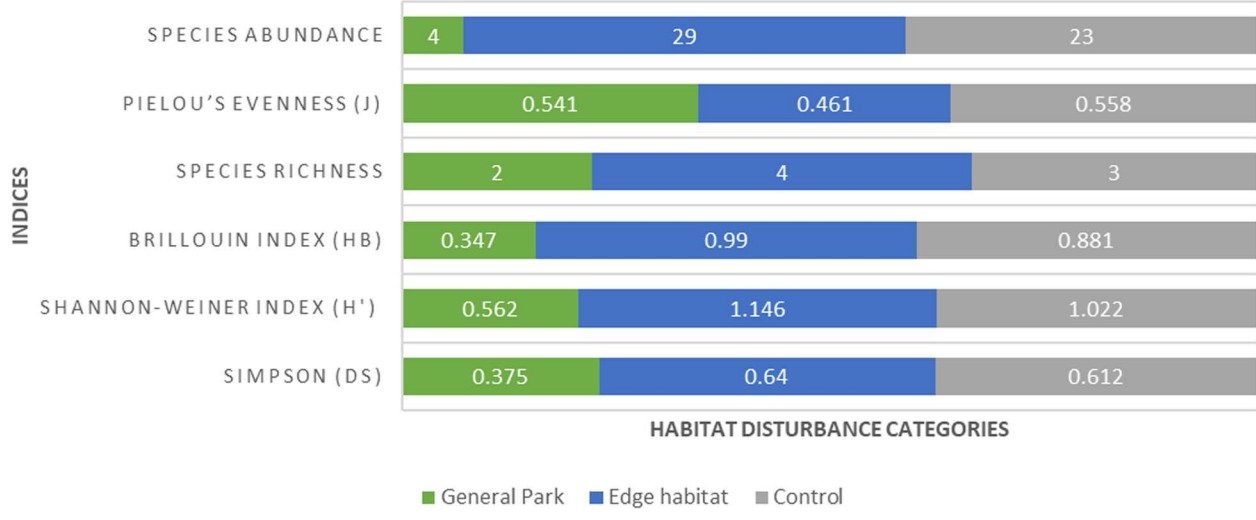

**Fig 3. Comparison of rodent diversity, species richness, evenness and abundance between the general park and edge habitats, including control in Nairobi National Park.**

## Habitat variability

The outcome from the Principal Component Analysis of tree and shrub densities and herbaceous cover from 15 sites belonging to the three vegetation classes resulted in three principal axes accounting for 100% of the variance, with the first two axes responsible for about 93% of the variance. On the other hand, PCA results based on the dominant plant species in each sampling site produced 15 principal axes from the presence and absence of 31 plant species. Twelve of these principal components explained 100% of the variance, with 6 principal components explaining 90% of the variance. Projecting vegetation types and habitat disturbance

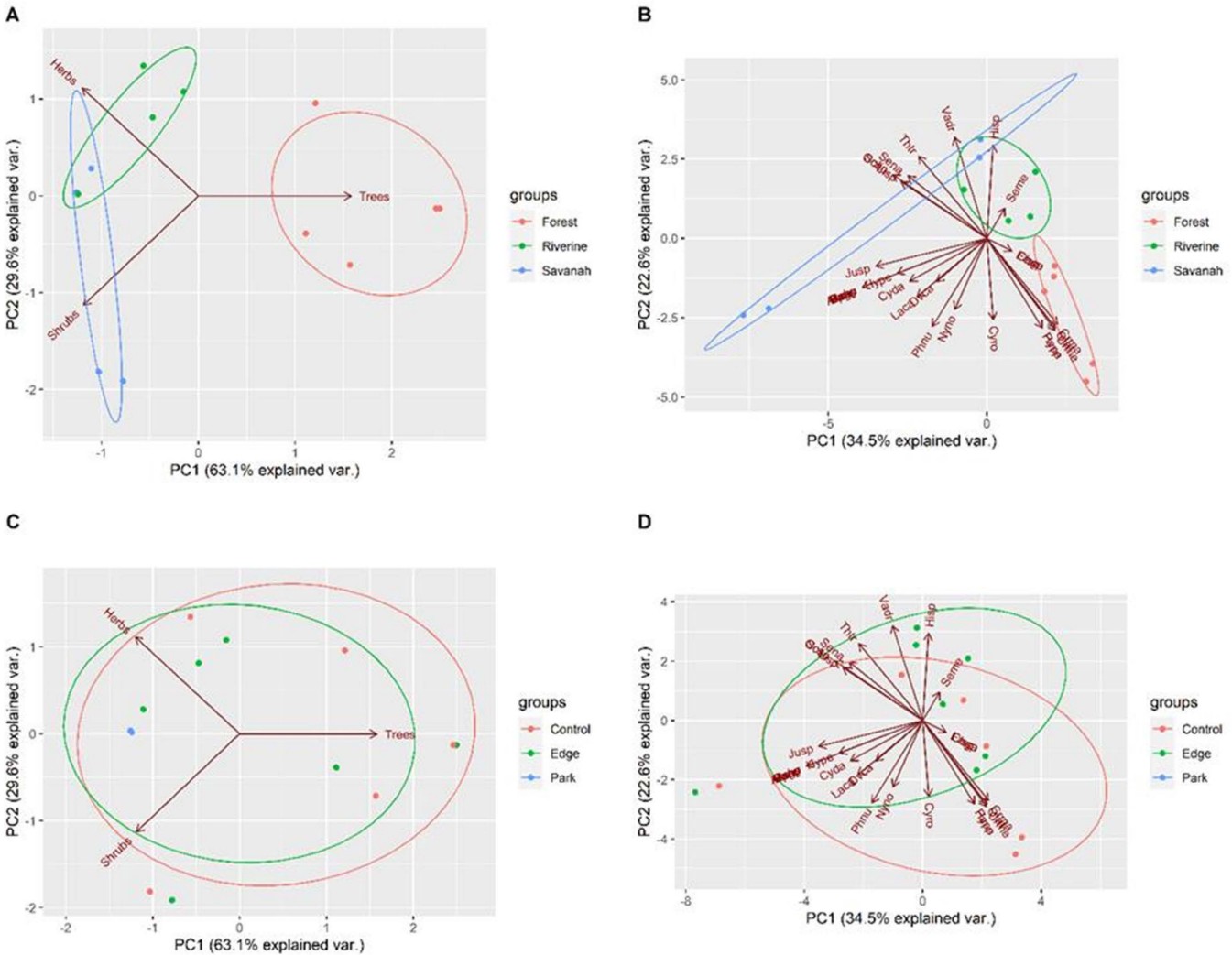

**Fig 4. Principal component analyses biplots showing clustering of vegetation type (A, B) and human disturbance (C, D) using shrub density, tree density, and herbaceous cover (A, C) and species composition (B, D).**

groupings on PCA axes revealed that vegetation types could be discriminated from habitat metrics (Fig 4A and 4B) but not human disturbance (Fig 4C and 4D).

Analyses of Variance revealed that tree (F = 28.95, p <.0001) and shrub mean densities (F = 10.17, p < 0.0026) were significantly different across vegetation types (Fig 5). However, variation was not statistically significant between disturbed areas and controls (Figs 5 and 6). Similarly, Kruskall-Wallis analyses also revealed that median herbaceous cover variations were statistically significant across vegetation types H(3) = 7.757, p < 0.0207 but not across human disturbance categories.

Herbaceous cover, shrub density and tree density were highly variable in control sites compared to disturbed sites, but the differences were not statistically significant (Fig 6A–6C). The herbaceous cover was higher in riverine and savannah vegetation compared to forests (Fig 6D). It was also noted that shrub density was highest in savannah and lowest in forest vegetation, but it was intermediate in riverine habitats (Fig 6E). In contrast, tree

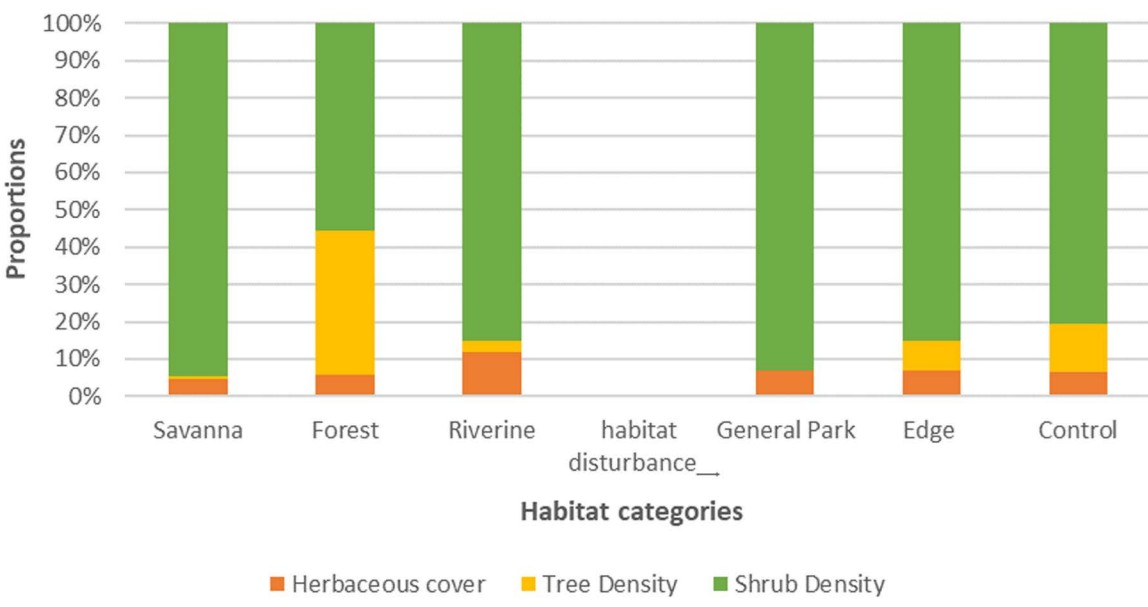

**Fig 5. The mean and median variation in shrub and tree density per hectare and percent herbaceous cover across vegetation types and habitat disturbance categories.**

density was highest in forest vegetation but was lower in savannah and riverine vegetation (Fig 6F).

Dominant plant species composition varied across vegetation types (Fig 7). The grass species (*Themeda trianda)* was the primary vegetation in all the sites in savannah. However, the dominant herb species in the savannah included *Lippia* sp., *Ocimum suave* and *Solanum incanum,* whereas the dominant trees were *Searsia natalensis*, and *Vachellia drepanolobium*.

In contrast to savannah, the forest vegetation was dominated by different tree species, including *Croton megalocarpus, C. macrostachys* and *Olea europaea subsp cuspidate.*

The species composition for the riverine vegetation somewhat overlapped with savannah vegetation. The primary grass species in the riverine was also *Themeda trianda*, whereas the dominant trees were *Searsia natalensis* and *Vachellia drepanolobium* (Fig 7). It was noted that some plants were restricted to particular vegetation types. The grasses restricted to the forest included *Panicum* sp., *Chloris gayana*, and *Eragrostis sp.* while the trees restricted to the forest included *Croton megalocarpus*, *Olea europaea subsp cuspidate* and *Croton macrostachyus*. Species restricted to savannah were *Vachellia gerrardii* and *Balanites aegyptiaca* among trees, *Opuntia* sp., and *Hyphaene* sp., among shrubs and *Asparagus* sp., *Parthenium hysterophorus* and *Aspilia mossambicensis* among herbs. No grass species was restricted into a savannah vegetation. *Senegalia mellifera* was the only tree restricted to riverine vegetation.

There was limited conspicuous differentiation by plant species composition across habitat disturbance categories. The dominant species in the control sample sites for human disturbance included *Themeda trianda* and *Cyperus rotundus* among grasses, *Phyllanthus nummulariifolius* var. *capillaris* among shrubs and *Searsia natalensis*, *Vachellia drepanolobium*, *Olea europaea subsp. cuspidate* and *Croton macrostachyus* among the trees.

Human-disturbed habitats (edges) also had *Themeda trianda* as dominant grass, *Lippia* spp as dominant herb and *Searsia natalensis* as dominant tree. The plants dominating the park were grass species *Themeda trianda*, herb species *Lippia* sp, *Ocimum suave, Solanum incanum*

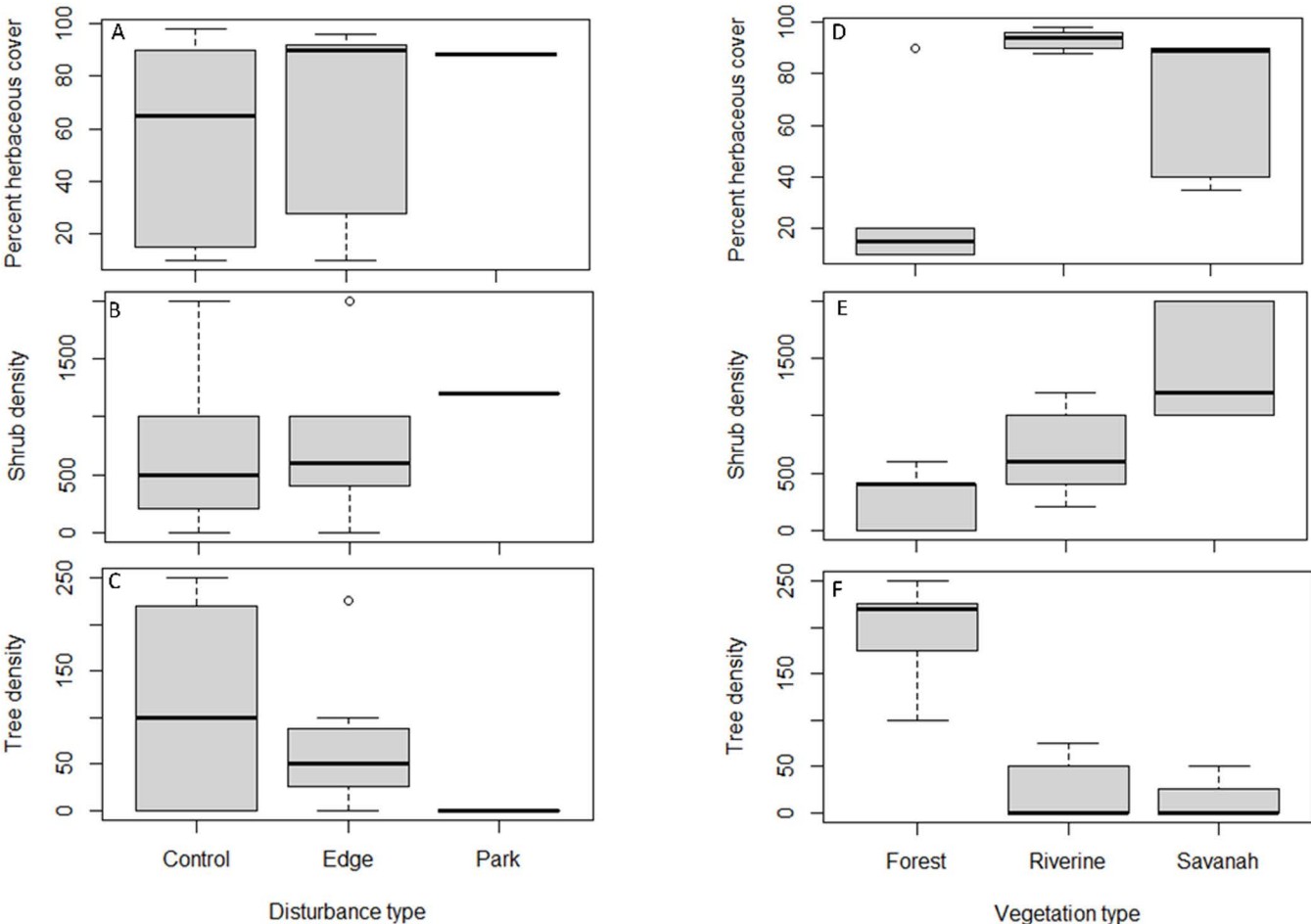

**Fig 6. Boxplots showing the variation in herbaceous cover, shrub density, and tree density with respect to habitat disturbance (A, B, C) and vegetation type (D, E, F) in NNP.**

and *Hibiscus* sp, while *Searsia natalensis* and *Vachellia drepanolobium* were among the dominant tree species in the park as a pristine category (Fig 7).

It was observed that 'control sites' for the 'edge habitats' had more restricted species such as *Cyperus rotundus*, *Chloris gayana*, *Eragrostis sp* among grasses, *Nymphaea nouchali*, *Hypenia* sp among the herbs and *Carissa spinarum* among the shrubs. *Hyphaene* sp, *Dovyalis caffra* and *Senegalia mellifera* were the plant species restricted to the edge or disturbed habitats (Fig 7).

**The influence of vegetation metrics, vegetation type, human disturbance, and seasonality on rodent abundance, species richness, and diversity.** Univariate Generalized Linear Models revealed that the abundance of rodents in this study was influenced by season, vegetation type, and vegetation metrics (density and cover) but not human disturbance (Table 1). The best multivariate model, however, indicated that rodents were more abundant in the wet season compared to the dry season, and abundance was also positively associated with increased tree and shrub densities (Table 1, S1 Table).

Rodent species richness was only positively associated with higher tree density (Table 2). Multivariate model selection indicated that there was no better model than a univariate tree density model (Table 2, S2 Table). The model, including vegetation type and intercept, was also a supported model explaining rodent species richness in NNP (S2 Table).

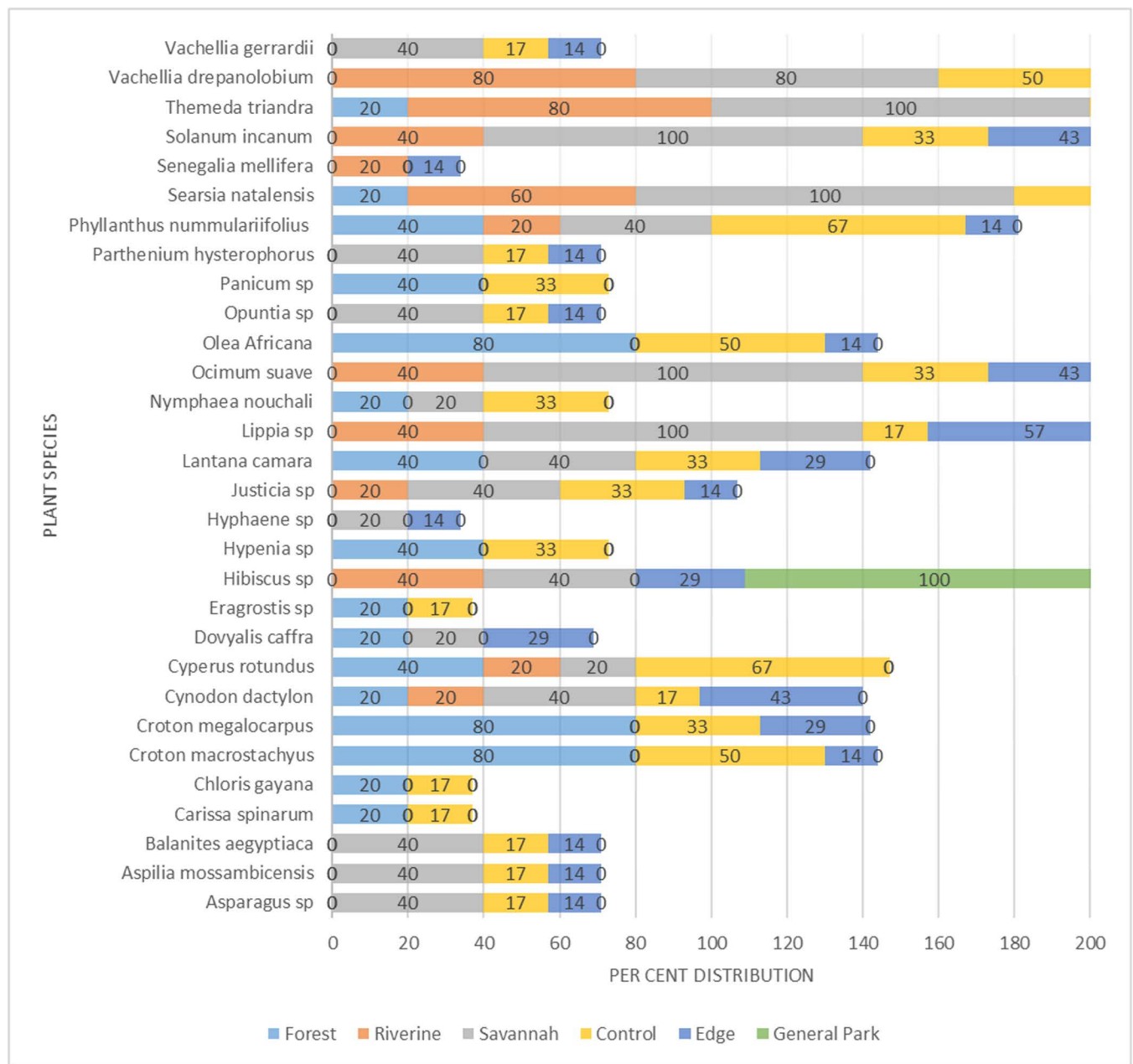

**Fig 7. The percent distribution of dominant plant species in sites sampled by vegetation type (forest, riverine and savannah,) and disturbance (edge and control).**

Univariate linear regression model selection using AIC revealed that the best model for species diversity estimated using the Shannon-Weiner index (Table 3), Brillouin index (Table 4) or Simpson index (Table 5) was vegetation type. Specifically, forest vegetation had higher species diversity than either savannah or riverine vegetation. On the other hand, the best over-all model selected after evaluating all simple and complex multivariable models (S3–S5 Tables) was a univariate model with diversity positively influenced by tree density (Table 3 and 4).

The human disturbance had no influence on diversity estimated using the Shannon-Weiner diversity index (Table 2), Brillouin index (Table 3) or Simpson index (Table 4).

**Table 1. Univariate models and the best multivariate model explaining rodent abundance in Nairobi National Park, Kenya.**

|  | Estimate | Std. error | Z value | PR(>\|Z\|) | AIC |
|---|---|---|---|---|---|
| Univariate models |  |  |  |  |  |
| Intercept | 0.385 | 0.163 | 2.356 | 0.0185 |  |
| Tree density | 0.667 | 0.128 | 5.208 | 0.0000 | 119.0 |
| Intercept | 1.253 | 0.169 | 7.411 | 0.0000 |  |
| Riverine cf. forest | -1.476 | 0.392 | -3.766 | 0.0002 |  |
| Savannah cf. forest | -0.990 | 0.325 | -3.049 | 0.0023 | 127.8 |
| Intercept | -0.143 | 0.277 | -0.516 | 0.6059 |  |
| Wet season cf. Dry season | 1.196 | 0.317 | 3.779 | 0.0002 | 129.9 |
| Intercept | 0.480 | 0.153 | 3.144 | 0.0017 |  |
| Herbaceous cover | -0.537 | 0.138 | -3.899 | 0.0001 | 130.9 |
| Intercept | 0.577 | 0.140 | 4.11 | 0.0000 |  |
| Shrub density | -0.325 | 0.154 | -2.109 | 0.0350 | 142.0 |
| Intercept | 0.651 | 0.209 | 3.12 | 0.0018 |  |
| Edge cf. Control | 0.078 | 0.279 | 0.278 | 0.7809 |  |
| Park cf. Control | -0.651 | 0.542 | -1.201 | 0.2298 | 146.6 |
| Best model |  |  |  |  |  |
| Intercept | -0.403 | 0.296 | -1.361 | 0.1736 |  |
| Wet season cf. Dry season | 1.168 | 0.317 | 3.69 | 0.0002 |  |
| Tree density | 0.873 | 0.184 | 4.732 | 0.0000 |  |
| Shrub density | 0.326 | 0.190 | 1.715 | 0.0864 | 104.0 |

**Table 2. Univariate models explaining the association between specific variables with rodent species richness in Nairobi National Park, Kenya.**

|  | Estimate | Std. error | Z value | PR(>\|Z\|) | AIC |
|---|---|---|---|---|---|
| Intercept | -0.348 | 0.225 | -1.550 | 0.1210 |  |
| Tree density | 0.397 | 0.195 | 2.037 | 0.0417 | 66 |
| Intercept | -0.331 | 0.222 | -1.493 | 0.1355 |  |
| Herbaceous cover | -0.362 | 0.208 | -1.740 | 0.0818 | 66.9 |
| Intercept | 0.182 | 0.289 | 0.632 | 0.5280 |  |
| Riverine cf. forest | -0.876 | 0.532 | -1.645 | 0.1000 |  |
| Savannah cf. forest | -0.693 | 0.500 | -1.386 | 0.1660 | 68.4 |
| Intercept | -0.511 | 0.333 | -1.532 | 0.1250 |  |
| Wet season cf. dry season | 0.442 | 0.427 | 1.034 | 0.3010 | 68.9 |
| Intercept | -0.271 | 0.210 | -1.294 | 0.1960 |  |
| Shrub density | -0.110 | 0.221 | -0.498 | 0.6190 | 69.7 |
| Intercept | -0.288 | 0.333 | -0.863 | 0.3880 |  |
| Edge cf. control | 0.047 | 0.450 | 0.104 | 0.9180 |  |
| Park cf. Control | 0.000 | 0.667 | 0.000 | 1.0000 | 72 |

## Discussion

### Patterns of rodent diversity (richness, evenness) and abundance by sampling site, vegetation type and human disturbance

In this study, five rodent species were detected, of which three are native (*L. striatus*, *Otomys tropicalis*, *Hylomyscus* sp) while two are invasive/exotic commensal species (*Mus mus,* and

**Table 3. Univariate models for Shannon-Weiner Index of rodent species diversity.**

|  | Estimate | Std. error | T value | PR(>|T|) | AIC |
|---|---|---|---|---|---|
| Intercept | 0.592 | 0.138 | 4.292 | 0.0026 |  |
| Riverine cf. forest | -0.592 | 0.195 | -3.035 | 0.0162 |  |
| Savannah cf. forest | -0.592 | 0.211 | -2.810 | 0.0229 | 7.4 |
| Intercept | 0.215 | 0.099 | 2.182 | 0.0570 |  |
| Tree density | 0.232 | 0.104 | 2.241 | 0.0518 | 10.4 |
| Intercept | 0.215 | 0.110 | 1.954 | 0.0825 |  |
| Shrub density | -0.173 | 0.116 | -1.497 | 0.1686 | 12.9 |
| Intercept | 0.215 | 0.120 | 1.802 | 0.1050 |  |
| Herbaceous cover | -0.094 | 0.125 | -0.748 | 0.4740 | 14.7 |
| Intercept | 0.234 | 0.208 | 1.126 | 0.2930 |  |
| Edge cf. control | 0.052 | 0.279 | 0.188 | 0.8560 |  |
| Park cf. control | -0.234 | 0.360 | -0.650 | 0.5340 | 16.4 |

**Table 4. Univariate models for Brillouin Index for rodent species diversity.**

|  | Estimate | Std. error | T value | Pr(>|t|) | AIC |
|---|---|---|---|---|---|
| Intercept | 0.422 | 0.094 | 4.484 | 0.0020 | -1.06 |
| Riverine cf. forest | -0.422 | 0.133 | -3.171 | 0.0132 |  |
| Savannah cf. forest | -0.422 | 0.144 | -2.936 | 0.0188 |  |
| Intercept | 0.153 | 0.066 | 2.336 | 0.0443 | 1.46 |
| Tree density | 0.176 | 0.069 | 2.553 | 0.0310 |  |
| Intercept | 0.153 | 0.076 | 2.009 | 0.0754 | 4.77 |
| Shrub density | -0.126 | 0.080 | -1.576 | 0.1494 |  |
| Intercept | 0.153 | 0.084 | 1.830 | 0.1010 | 6.83 |
| Herbaceous cover | -0.063 | 0.088 | -0.722 | 0.4890 |  |
| Intercept | 0.173 | 0.145 | 1.189 | 0.2680 | 8.53 |
| Edge cf. control | 0.026 | 0.195 | 0.134 | 0.8970 |  |
| Park cf. control | -0.173 | 0.252 | -0.687 | 0.5120 |  |

**Table 5. Univariate models for the Simpson index of rodent species diversity.**

|  | Estimate | Std. error | T value | Pr(>|T|) | AIC |
|---|---|---|---|---|---|
| Intercept | 0.358 | 0.082 | 4.361 | 0.0024 | -4.0 |
| Riverine cf. forest | -0.358 | 0.116 | -3.083 | 0.0150 |  |
| Savannah cf. forest | -0.358 | 0.125 | -2.855 | 0.0213 |  |
| Intercept | 0.130 | 0.059 | 2.222 | 0.0534 | -1.0 |
| Tree density | 0.142 | 0.061 | 2.312 | 0.0460 |  |
| Intercept | 0.130 | 0.066 | 1.973 | 0.0799 | 1.6 |
| Shrub density | -0.105 | 0.069 | -1.522 | 0.1623 |  |
| Intercept | 0.130 | 0.072 | 1.814 | 0.1030 | 3.4 |
| Herbaceous cover | -0.057 | 0.075 | -0.754 | 0.4700 |  |
| Intercept | 0.142 | 0.125 | 1.137 | 0.2880 | 5.2 |
| Edge cf. control | 0.031 | 0.168 | 0.185 | 0.8580 |  |
| Park cf. control | -0.142 | 0.216 | -0.657 | 0.5300 |  |

*Rattus* sp). The dominance of *L. striatus* (striped grass mouse) in NNP with 43.9% (25/57) suggests its adaptability to the current ecological state of the park. Specifically, it was noted that *L. striatus* occurred across the three primary vegetation types in NNP, while the Pielou's species evenness was moderate (0.44), indicating nearly average equity in species distribution. The *L. striatus* is one of the common rodents in grasslands, savannah, and cultivated areas, with reports showing that it has adapted to disturbed habitats in the tropics [36,37]. The degree to which human interference changes habitat structure, and the extent to which infrastructural developments occur in the landscape may affect several ecological factors [38]. Human activities, including infrastructure projects such as railway lines and road networks, recreational places, and the rising levels of air, water and noise pollution in and around NNP, account for significant ecosystem disturbance. Habitat disturbance that modifies habitat structure can affect small mammal populations [38] and may result in local extirpation of rodent species, especially in urbanized habitats [39,40].

Considering that there is no documentation on the previous diversity of rodent species in the NNP for comparison with our findings, it is sufficient to conclude that *L. striatus, O. tropicalis,* and *Hylomyscus* sp. are the natural remnant species of the NNP. Only *Crocidura fumosa* and *Mastomys coucha* were caught in four, 320-night traps at Embakasi Plains, the only place with a report on rodent species close to NNP [41]. Because the genus *Mastomys* is found throughout Africa, with *M. natalensis* being most common in East and Central Africa, its absence in NNP was unexpected and may imply local population extinction.

The genus *Hylomyscus* (wood mice) are montane specialists, and were not expected in NNP. *H. endorobae* and *H. kerbispeterhansi* occur in tropical African lowland and montane rainforests, including Mt. Kenya and Mt. Elgon [42–44], while *H. denniae* are endemic to the Mau escarpment in Kenya [45]. Perhaps the *Hylomyscus* sp. in NNP could be a distinct species, given that recent surveys and genetics have revealed more species within the genus [46].

The rodent species richness in NNP was comparable to the Laikipia rangeland, Kenya, where richness ranged from 5–7 [47]. In contrast, species richness is higher in some other African savannah landscapes. In Katavi and Mikumi National Parks, both in Tanzania, five and 21 species were observed [36,48], while Nechisar National Park, Ethiopia, had 20 species [49]. In the aforementioned studies, higher rodent abundance and diversity were observed in human-settled sites or disturbed/edge habitats. Since most of the human-occupied sites were in the forested areas of NNP and because forest vegetation types had higher rodent diversity, richness, and abundance than savannah vegetation types, both vegetation type and human disturbance likely affected rodent population metrics. According to Jeffrey (1977), clearing forests and replacing the patches with domestic housing and cultivation increases the diversity and abundance of rodents [50]. Generally, forest vegetation hosts higher rodent diversity than savannah. Olayemi and Akinpelu (2008) observed a similar pattern in Nigeria, where Shannon-Weiner Index was higher in the forest (H = 1.68) compared to the derived Savannah (H = 0.97) [51]. This pattern is consistent with observations at Mikumi National Park (Tanzania), Bwindi Impenetrable National Park and Kibale National Park (Uganda), where the evergreen forest showed the highest species diversity, compared to the Savanna woodlands [48,52,53]. It is posited that evergreen forests support a variety of food resources and provide several microhabitats which may offer cover and nest sites to different small mammal species [52,53].

**Factors driving rodent abundance, species richness and diversity.**  Multiple biophysical factors, such as predator risks and avoidance opportunities, intra-and inter-species competitions, resource quantity and quality, and especially the availability of water and food, influence rodent community composition, richness, abundance and diversity [54]. In NNP,

rodent abundance was influenced by season, vegetation type and vegetation metrics (density and cover).

The rodents were more abundant in the wet season compared to the dry season, which is a pattern consistent with African rodent communities [55]. Availability of rainfall is highly correlated with the reproduction of small mammals in Africa [56,57], particularly with the breeding and population dynamics of several rodent species [58]. It was also established that abundance was positively associated with density and cover, especially tree and shrub density in NNP. The spatial distribution of rodents in a landscape is determined by intrinsic and ecological factors, as each species responds differently to habitat structural variability [59]. Rainfall and habitat productivity are interdependent and drive rodent abundance and richness in NNP. Rainfall may directly influence vegetation height and density, hence the resulting cover, which are components of suitable habitats for rodents [55,60]. Less vegetation cover increases predation risk, reduces food quality, and promotes negative competition, eventually affecting population performance [61,62].

Vegetation type significantly influences rodent species diversity, as supported by the Shannon-Weiner index, Brillouin index, and Simpson index analyses. Specifically, vegetation type determines the availability of different resources for instance food, water and shelter needed by different rodents, impacting their abundance and diversity. Our findings revealed that as tree density increased, so did rodent variety in NNP, and higher tree density is frequently associated with forests [63]. The high canopy tree cover may provide safety within a predator-rich habitat like the NNP or species-specific microclimate (cool and humid) preferences and a wider food range [29]. Moreover, in the present study, most of the human settlements were located within the forested part of NNP, and the settlements were associated with the invasive *Rattus* and *Mus* species, hence resulting in the overall increase of diversity in the forest area. Less species diversity was expected in the cleared patches that are used for human settlements and infrastructure development in the NNP [64], but instead, these disturbances had little effect on rodent diversity in the park. The findings of this study distinguish between the influence of anthropogenic activities and the effect of the habitat type in which such human activities occur.

In conclusion, the vegetation types influenced rodent diversity, while species richness was only positively associated with higher tree density. Season, vegetation type, and vegetation metrics (density and cover) all influenced rodent abundance. Anthropogenic habitat disturbance has no direct influence on rodent abundance. A high forest density and coverage may favor a higher diversity of rodent species.

## Supporting information

**S1 Table. Model selection table for species abundance.**
(DOCX)

**S2 Table. Model selection table for rodent species richness.**
(DOCX)

**S3 Table. Model selection table for Shannon Weiner diversity Index.**
(DOCX)

**S4 Table. Model selection table Brillouin index.**
(DOCX)

**S5 Table. Model selection table for Simpson's diversity Index.**
(DOCX)

## Acknowledgments

We appreciate the National Museums of Kenya and the Kenya Wildlife Service for all the logistical support. We also appreciate the research assistants and security rangers for their assistance during fieldwork, especially Mr. Daniel Muteti of Kenya Wildlife Service and the sampling team to the Nairobi National Park while Mr. Jackson King'oo and Mr. Elphas Bitok facilitated the fieldwork. Sincerely appreciation for the logistic and ranger security support from Sergents Evans Ochieng, Joseph Thoya, John Nderitu, and Paul Ngechu (KWS) during field visits to the study sites in Nairobi National Park.

## Author contributions

**Conceptualization:** Immaculate M. Mungai, Nathan Gichuki, Benard Agwanda, Olivia Wesula Lwande.

**Data curation:** Immaculate M. Mungai, Benard Agwanda, Vincent Obanda.

**Formal analysis:** Immaculate M. Mungai, Patrick Chiyo.

**Funding acquisition:** Olivia Wesula Lwande.

**Investigation:** Benard Agwanda, Vincent Obanda.

**Methodology:** Immaculate M. Mungai, Benard Agwanda, Vincent Obanda.

**Project administration:** Olivia Wesula Lwande.

**Resources:** Olivia Wesula Lwande.

**Supervision:** Vincent Obanda.

**Validation:** Nathan Gichuki, Vincent Obanda, Olivia Wesula Lwande.

**Visualization:** Olivia Wesula Lwande.

**Writing – original draft:** Immaculate M. Mungai, Nathan Gichuki, Dorcus A.O. Sigana, Vincent Obanda, Olivia Wesula Lwande.

**Writing – review & editing:** Immaculate M. Mungai, Nathan Gichuki, Dorcus A.O. Sigana, Vincent Obanda, Olivia Wesula Lwande.

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
