## [Decision Letter · Decision Letter 0]

2 Oct 2023

PONE-D-23-23260Drivers of Rodent Community Structure in an Urban National Park, KenyaPLOS ONE

Dear Dr. Lwande,

Thank you for submitting your manuscript to PLOS ONE. After careful consideration, we feel that it has merit but does not fully meet PLOS ONE’s publication criteria as it currently stands. Therefore, we invite you to submit a revised version of the manuscript that addresses the points raised during the review process.

The authors must address the comment as there are issues related to the calculation of indices and language. 

We look forward to receiving your revised manuscript.

Kind regards,

Lalit Kumar Sharma

Academic Editor

PLOS ONE

Journal Requirements:

5. We note that Figure 1 in your submission contain map/satellite images which may be copyrighted. All PLOS content is published under the Creative Commons Attribution License (CC BY 4.0), which means that the manuscript, images, and Supporting Information files will be freely available online, and any third party is permitted to access, download, copy, distribute, and use these materials in any way, even commercially, with proper attribution. For these reasons, we cannot publish previously copyrighted maps or satellite images created using proprietary data, such as Google software (Google Maps, Street View, and Earth). For more information, see our copyright guidelines: http://journals.plos.org/plosone/s/licenses-and-copyright.

Additional Editor Comments:

The authors must address the comment as there are issues related to the calculation of indices and language. The manuscript is overall week in terms of readability and lack clarity. Further, the manuscript is not prepared as per the PLOS ONE format; at least the primary test setting should be done so that the reviewer can follow the text.

Reviewers' comments:

Reviewer's Responses to Questions

**Comments to the Author**

1. Is the manuscript technically sound, and do the data support the conclusions?

Reviewer #1: Yes

Reviewer #2: Partly

2. Has the statistical analysis been performed appropriately and rigorously? 

Reviewer #1: Yes

Reviewer #2: Yes

3. Have the authors made all data underlying the findings in their manuscript fully available?

Reviewer #1: Yes

Reviewer #2: Yes

4. Is the manuscript presented in an intelligible fashion and written in standard English?

Reviewer #1: Yes

Reviewer #2: No

5. Review Comments to the Author

Reviewer #1: The authors need to revise the ms throughly. I won't emphasize on language but there are sentences which doesn't make sense. I have provided my comments as an attachment. Please find the pdf attached and revise the ms accordingly.

Reviewer #2: The current research holds significant importance as it offers insights into a frequently overlooked category of organisms, namely, small mammals. Below are my overall remarks on the manuscript, with more detailed comments provided in the PDF document.

1. Abstract: The abstract requires certain adjustments in terms of its writing flow and cohesion. Some of the statements in the abstract appear incomplete. Specifically, it lacks information regarding the statistical methods employed to achieve the research objectives.

2. Introduction: The introduction is competently written, but it appears somewhat disjointed and lacks cohesiveness in its narrative. To enhance its clarity, the authors should initially provide a clear definition of the term "Urban protected area." Following this, the first paragraph should incorporate substantial ecological insights into rodents and their pivotal role in maintaining ecosystem health. For instance, rodents serve as crucial prey for meso and small predators, as well as seed predators. The introduction should then delve into how human disturbances and alterations could impact rodent populations and diversity, leading to a discussion of conservation concerns within the National Park. Lastly, it would strengthen the manuscript's scientific rigor to include a hypothesis and corresponding predictions in the conclusion of the introduction.

In the nutshell authors need to make strong ecological rationale with unity and flow in writing.

3. Methods:

Rodent community survey

The Methods section appears disorganized and lacks a clear direction, leading to confusion for readers. Upon closer examination, it seems that the authors placed transects in various habitats and subsequently employed Sherman trapping. However, the current presentation makes it challenging to comprehend the step-by-step process of their sampling, from habitat selection to rodent capture. To improve clarity, the authors should simplify and provide more detailed explanations of their survey methods, ensuring that the sequence is easy for readers to follow. Additionally, statements like "edge effects" need further elaboration to enhance understanding.

Tree and Shrub Density and their Percent Cover

This section should be consolidated under a single heading, such as "Vegetation Variables," as the scattered information under different headings lacks coherence. Additionally, it remains unclear how the quadrates were positioned on the transects—whether they were placed along the transects or centered on Sherman traps. This ambiguity raises concerns regarding the representation of fine-scale rodent habitat usage, which could have substantial implications for the results and their interpretation.

Furthermore, statements like "A 30 x 30cm wire quadrat, thrown along each transect" in line 153 contribute to the confusion in the sampling design.

Moreover, there is insufficient detail provided on how data related to anthropogenic variables was collected during the sampling process. Crucially, the manuscript does not specify how rodent abundance was assessed, whether through traditional capture-recapture methods as briefly mentioned or through alternative methodologies. Clarifications on these points would enhance the comprehensibility of the methodology section and the study as a whole.

Data analysis

Rodent Species Diversity, Richness, Evenness and Abundance

he rationale behind the inclusion of numerous diversity indices needs clarification in the manuscript. Additionally, there is a lack of information regarding how the richness and evenness indices were calculated. Furthermore, the calculation method for abundance is not clearly stated.

To improve the clarity and conciseness of this section, it would be beneficial for the authors to provide explanations for why they chose to use multiple diversity indices, offer details on how richness and evenness indices were computed, and specify the method employed for calculating abundance. Simplifying the presentation of this information will help readers better understand the study's methodology.

Tree and Shrub Density and their Percent Cover

The manuscript would benefit from a clearer rationale for the inclusion of PCA analysis in investigating vegetation differences in various habitats, as it currently appears to create a disconnect between vegetation structure and factors influencing rodent diversity. Staying closely aligned with the study objectives would improve the manuscript's focus. Additionally, it is important for the authors to support their statements with references, especially in the statistical section, to enhance the scientific rigor of their work. The statistical section should not read like a storytelling or blog but should be underpinned by relevant literature and methodology.

Regarding the use of both bivariate and multivariate predictive modeling, the authors should provide a strong justification for this choice. It would be helpful to clarify why they employed both approaches and how they complement each other, especially in cases where GLM with combinations of models may be more powerful. A clear rationale will provide readers with insight into the methodology selection and strengthen the scientific basis of the analysis.

Results

The Results section should be rewritten for improved clarity. It's essential to provide a clear explanation of the type or species groupings that the PCA has undertaken.

Furthermore, considering that the main focus of the manuscript is on rodents, it might be best to maintain consistency in the treatment of the vegetation section. Deviating from the core focus on rodents can create distractions, so it's advisable to align the presentation with the central theme of the study.

Discussion.

Need more strengthening from the perspective of rodent ecology.

In summary, the manuscript exhibits issues with unity and writing flow, resulting in a fragmented understanding of its sections. There is a perception that the authors have introduced unnecessary objectives related to vegetation and its statistical analysis, which detracts from the central focus on rodents. To enhance the manuscript's potential for publication in PLOS ONE, it is advisable to concentrate solely on the rodent aspect and consider restructuring the manuscript for better scientific coherence. A recent manuscript by Misher et al. (2022) on a similar topic could serve as a helpful reference for organizing the manuscript more effectively. A major revision is recommended to address these concerns and improve the overall quality of the manuscript.

Misher C, Vats G and Vanak AT (2022) Differential Responses of Small Mammals to Woody Encroachment in a Semi-Arid Grassland. Front. Ecol. Evol. 10:755903. doi: 10.3389/fevo.2022.755903

6. PLOS authors have the option to publish the peer review history of their article (what does this mean? ). If published, this will include your full peer review and any attached files.

**Do you want your identity to be public for this peer review?** For information about this choice, including consent withdrawal, please see our Privacy Policy .

Reviewer #1: **Yes: ** Amira Sharief

Reviewer #2: No

---

## [Author Response · Author response to Decision Letter 1]

11 Jan 2024

29th November 2023

Editor-in-Chief,

Emily Chenette (PhD),

PLOS One.

Dear Emily Chenette,

We are happy to resubmit this research article entitled “Drivers of Rodent Community Structure in an Urban National Park, Kenya- Manuscript number- PONE-D-23-23260” by Immaculate M. Mungai, Nathan Gichuki, Dorcus A.O. Sigana, Benard Agwanda, Patrick Chiyo, Vincent Obanda, Olivia Wesula Lwande to the PLOS One Journal.

We are grateful for the valuable feedback from the reviewers. The comments raised by all reviewers are vital to us - to improve the quality of this manuscript, and for expanding the understanding, not only on how urbanization affects preservation of biodiversity, but also the influence of anthropogenic activities and the effect of the habitat type in which such human activities occur and how these affects the rodent structure.

We hope that our revised manuscript will illuminate some of these important questions better and will be considered for publication in the PLOS One Journal.

Please find the point-by-point responses below (in red highlight).

On behalf of all authors,

Olivia Wesula Lwande, corresponding author,

Department of Clinical Microbiology,

901-85 SE, University of Umeå, Sweden

olivia.lwande@umu.se

To the Editor, thank you for the extensive and meaningful comments. Find below the reply to the comments raised.

1.The manuscript has been revised according to PLOS ONE’s formatting style and file names.

2.Language, including grammar and spelling has been checked and revised as appropriate by Dr. George O. Paul (gomondip@gmail.com ) with help from the EditScope language check and editing services provider.

Tracked manuscript included – *supporting Information* file

Clean copy of the edited manuscript – new *manuscript* file

3.Funding-related text has been deleted from the Manuscript.

4.Ethics statement have been moved to the Methods section.

5.Figure 1 is a map that was created by the author and NOT sourced from a copyright owner.

6.Captions for supporting files have been placed at the end of the manuscript and text citations and file names updated as per the format.

Reviewer 1

Line 31: Sentence revised as follows: Nairobi National Park (NNP) is among the most vulnerable ecosystems.

Line 40: The abbreviation NNP is now explained in line 31.

Line 127: The sentence has been revised accordingly.

Line 140: We have given examples of some species-specific traits that may be used in rodent identification in the field. Identification of rodents in the field is based on morphological traits. There are several traits or characteristics that are specific to a species. For instance, some species have hair, while others have fur and others are naked like the naked-mole rats. The hair or fur may be of different colouration/ patterning; The tail may be shorter than the body or longer than the body.

Line 142: We have revised the title as suggested ‘Vegetation Sampling’.

Line 143 -154: Following the change above, and for clarity and readability, this section has been revised.

Line 168 -175: The section has been revised as suggested by the reviewer.

Line 187: Deleted and redrafted in line line 176-177 under Shannon Wieiner index.

Line 206-212: The sentence has been revised.

Line 222-223: Citation has been included.

Line 235-236: Citation has been included.

Reviewer 2

Abstract:

Line 40-46: describes the statistics used in the study, such as the diversity indices, GLM and both univariate and multivariate models have been mentioned.

Introduction: This section has been revised to ensure cohesiveness and flow as guided by the reviewer in their editorial notes and also in the tracked manuscript.

Line 56-57: We have described the Urban Protected area/ National Park and also provided a reference.

Line 57-66: For clarity and flow, we have revised the sentence, combining multiple sentences into one, and providing references.

Line 68-69: Sentence revised.

Line 73-74: Revised by merging with the preceding sentence in line 72-73.

Line 76-77: Sentence revised and reference added.

Line 79-81: Following the major revision, this section is removed.

Line 98: Corrections made as suggested by the reviewer.

Line 101-104: The sentence has been transferred to the Introduction.

Line 104- 106: Rainfall amount has been included in the sentence and also a citation.

Line 114-116: The sentence has been revised – replacing wildlife diversity with ‘Mammals’.

Methods

The supplementary data on species abundance are in S1 Table 1 species abundance.

The comments on capture-mark-recapture have been addressed.

Line 127 – 129: The sentence has been revised for clarity.

Line 129-130: We have provided elaboration and included additional citations.

Line 130-134: The sentences have been revised for clarity.

Line 144 -145: Citation format has been corrected according to the journal requirement.

Line 144-145: We have clarified that during sampling, the quadrats were within transects.

Line 153: We have clarified thrown along the transect to ‘thrown over the rodent trapping transects’. Since rodents were captured along specific transects, we wanted to determine the ground cover in the area within the transects.

Line 163-164: We have revised this sentence to show that we used three diversity indices and explained their purpose.

Line 198: The title has been revised – The density of trees and shrubs and herbaceous per cent cover.

Line 206-212: We have included citations.

Line 214-215: We tested differences in densities of trees and shrubs in Savanna, riverine and forest parts of NNP because these parameters may have an association with rodent community (Madden et al, 2019). Our results found that abundance was positively associated with density and herbaceous cover, especially tree and shrub density in NNP.

Line 217: The word ‘core’ has been deleted.

Line 220-223: Sentence revised, and citations included.

Line 241: The sentence has been revised for clarity - ‘self- triggered’ corrected as traps that triggered themselves but failed to capture.

Line 255 -263: These names are sampling locations, which have been provided in Table 1 and their locations are shown in the map (figure 1).

Line 363-373: Transferred to methods and placed under a sub-title – ‘Sampling sites’.

Line 294-313: We have used analysis to demonstrate how the habitat in NNP is variable. The objective of the study is to evaluate the relationship of habitat variability with rodent community structure. We consider this section to be a huge component of the study and if omitted will affect the objective of the study.

Line 315-316: We included Median values, besides the mean, because our large dataset had outliers that might skew the average of the values.

Line 391-393: We have mentioned this analysis in the Data analysis section line 232-233.

Line 424: The sentence has been revised and the word ‘Physical things’ – omitted.

Line 484-490: The discussion on this section have been improved as tracked.

---

## [Decision Letter · Decision Letter 1]

16 Jul 2024

PONE-D-23-23260R1Drivers of Rodent Community Structure in an Urban National Park, KenyaPLOS ONE

Dear Dr. Lwande,

Thank you for submitting your manuscript to PLOS ONE. After careful consideration, we feel that it has merit but does not fully meet PLOS ONE’s publication criteria as it currently stands. Therefore, we invite you to submit a revised version of the manuscript that addresses the points raised during the review process.

We look forward to receiving your revised manuscript.

Kind regards,

Dereje Yazezew Mammo, Ph.D.

Academic Editor

PLOS ONE

Journal Requirements:

Additional Editor Comments:

Thank you for considering PLOS ONE for your manuscript "Drivers of Rodent Community Structure in an Urban National Park, Kenya". Peer review of your manuscript is now complete. Based on the reports and my own reading of your manuscript, I have completed my evaluation of your manuscript.

Although the manuscript is thoughtful in its content and important for conservation endeavor, there are some areas that should be improved prior to publication as indicated by the reviewers. Especially the number of tables should be minimized by substituting with figures as per the feasibility. The references citations and bibliography sections also need revision, for instance the authors used both numbering and author in the citation and the journal names are written inconsistently with a mix of abbreviate and full written. There are also some discursive and self- contradictory run on sentences in the literature and discussion section without logical sequencing, lending to distractions and deviations from the key messages that the authors are trying to convey.

Reviewers' comments:

Reviewer's Responses to Questions

**Comments to the Author**

1. If the authors have adequately addressed your comments raised in a previous round of review and you feel that this manuscript is now acceptable for publication, you may indicate that here to bypass the “Comments to the Author” section, enter your conflict of interest statement in the “Confidential to Editor” section, and submit your "Accept" recommendation.

Reviewer #2: All comments have been addressed

Reviewer #3: (No Response)

2. Is the manuscript technically sound, and do the data support the conclusions?

Reviewer #2: Yes

Reviewer #3: Yes

3. Has the statistical analysis been performed appropriately and rigorously? 

Reviewer #2: Yes

Reviewer #3: Yes

4. Have the authors made all data underlying the findings in their manuscript fully available?

Reviewer #2: Yes

Reviewer #3: Yes

5. Is the manuscript presented in an intelligible fashion and written in standard English?

Reviewer #2: Yes

Reviewer #3: Yes

6. Review Comments to the Author

Reviewer #2: This work addresses the often overlooked rodents in comparison to their larger counterparts, highlighting its significance. In the initial manuscript review, I identified several weaknesses and issues, which I outlined. In the current review, the authors have effectively addressed all of my comments, making a compelling case for publication.

Reviewer #3: Dear authors,

Congratulations on the article.

From what I have seen, the manuscript is much improved since the first submission.

However, I think the abstract can be improved and I think tables are a bit excessive in the text.

You could transform tables into nice graphs, easier to visualise and compare results (Table 2).

Your figure 1 can hardly be understood by someone who does not know the study area, it should be improved.

In general, all figures and tables could be made more attractive and easier to visualise. The fact that most tables are desformatted makes interpretation difficult.

Overall, these are minor changes that can improve your manuscript and then I believe it is ready for publication.

Best wishes

7. PLOS authors have the option to publish the peer review history of their article (what does this mean? ). If published, this will include your full peer review and any attached files.

**Do you want your identity to be public for this peer review?** For information about this choice, including consent withdrawal, please see our Privacy Policy .

Reviewer #2: No

Reviewer #3: No

---

## [Author Response · Author response to Decision Letter 2]

10 Aug 2024

Nairobi National Park (NNP) is among Kenya's most vulnerable ecosystems, experiencing significant pressure from urbanization. Rodents, which are sensitive to environmental changes, are considered bioindicators of ecosystem health, and their population dynamics can be used to assess ecosystem pressures such as urbanization. This study assessed the rodent community structure in NNP to understand the effects of various urban pressures by examining the relationships between rodent diversity, richness, and abundance with vegetation types and metrics, seasonality, and habitat disturbances. The capture-mark-release method was used to trap rodents from 15 sites in Nairobi National Park's savannah, forest, and riverine vegetation types during the dry and wet seasons. The diversity, species richness and abundance were determined from the trappings. From 56 rodents trapped, five species identified were Lemniscomys striatus, Hylomyscus sp, Rattus rattus, Mus mus and Otomys tropicalis. Rodent diversity at NNP was low (Simpson=0.7130; Shannon Weiner=1.40; Brillouin index=1.27) while Pielou’s species evenness, was moderate=0.44 indicating near equity in species distribution. Univariate Generalised linear models showed that rodent abundance was influenced by season, vegetation type, and vegetation metrics but not by human disturbance/edge effects. The multivariate model indicated that rodents were more abundant in the wet season compared to the dry season, and that abundance was also positively associated with increased tree and shrub densities. Rodent species richness was positively associated with higher tree density, while vegetation types influenced rodent species diversity. Rodent abundance was influenced by vegetation type, vegetation metrics (density and cover), and season but not by human disturbance. It was observed that the diverse anthropogenic activities occurring in NNP, do not significantly influence rodent abundance compared to the biotic and abiotic factors. Factors that influence vegetation and season, are thus a threat to the faunal diversity in NNP. This first rodent survey in this Park provides preliminary data for continued monitoring of this ecosystem.

---

## [Decision Letter · Decision Letter 2]

3 Sep 2024

PONE-D-23-23260R2Drivers of Rodent Community Structure in an Urban National Park, KenyaPLOS ONE

Dear Dr. Lwande,

Thank you for submitting your manuscript to PLOS ONE. After careful consideration, we feel that it has merit but does not fully meet PLOS ONE’s publication criteria as it currently stands. Therefore, we invite you to submit a revised version of the manuscript that addresses the points raised during the review process.

We look forward to receiving your revised manuscript.

Kind regards,

Waheed Ali Panhwar, Ph.D

Academic Editor

PLOS ONE

Journal Requirements:

Reviewers' comments:

Reviewer's Responses to Questions

**Comments to the Author**

1. If the authors have adequately addressed your comments raised in a previous round of review and you feel that this manuscript is now acceptable for publication, you may indicate that here to bypass the “Comments to the Author” section, enter your conflict of interest statement in the “Confidential to Editor” section, and submit your "Accept" recommendation.

Reviewer #4: (No Response)

Reviewer #5: (No Response)

2. Is the manuscript technically sound, and do the data support the conclusions?

Reviewer #4: Yes

Reviewer #5: Yes

3. Has the statistical analysis been performed appropriately and rigorously? 

Reviewer #4: Yes

Reviewer #5: Yes

4. Have the authors made all data underlying the findings in their manuscript fully available?

Reviewer #4: Yes

Reviewer #5: Yes

5. Is the manuscript presented in an intelligible fashion and written in standard English?

Reviewer #4: Yes

Reviewer #5: Yes

6. Review Comments to the Author

Reviewer #4: This manuscript provides novel research into the significance of ongoing monitoring of ecosystems which are disturbed by human development and activities, with results applicable to land managers. While the manuscript generally reads well, it could improve through the addition of some fine-scale detail and further explanation of the results and their implications. Furthermore, incorporating the key findings in relation to effects on rodent diversity and abundance into the concluding remarks will ensure that readers grasp the main findings of the study clearly. There are also formatting inconsistencies through the manuscript which require correction to enhance the readability of the paper. The referencing, both in text and in the reference list, still needs some attention to adhere to the PLOS ONE referencing guidelines.

Detailed comments can be found below.

- Appears there may be extra spaces in the text between words/after full stops e.g. Lines 35, 70, 99, 105, 111, 173, 203, 210, 251, 335 (space before the comma), 404.

- Line 36: may read better as “were identified:”

- Lines 41, 46: instead of including that human disturbance had no effect on the two separate metrics, could have a sentence after stating what did have an effect on the metrics stating that human disturbance had no effect in both models.

- Line 48: add “measured” before “biotic and abiotic factors” to limit observations to those factors that were investigated in this study.

- Line 49: “a threat to the faunal diversity” – can the authors truly extrapolate their findings to all faunal diversity in the NNP with the same relationships they found for rodents occurring for all species?

- Line 54: change the full stop after the citation to a comma.

- Line 59: what frequency of monitoring is required? Ongoing?

- Line 62: outline what “the choice” is referring to e.g. the choice of what?

- Line 64: remove the “a” before survival.

- Lines 66, 68, 84, 91: inconsistent formatting of “e.g.” (e.g e.g., and e,g.,) – ensure all are formatted the same throughout the text.

- Line 68: remove the “;” after “prey”.

- Lines 87, 89, 107: the citations have circular brackets instead of square brackets.

- Line 92: is it better to say “park boundaries”?

- Line 98: start the sentence with “The”.

- Line 104: add “a” before “decline”.

- Line 110: “grass species, and scattered low-canopy” text shouldn’t be in italics.

- Line113-114 and 224-225: consider keeping format the same in relation to the use of “and” in the list of animals within the two lots of brackets (e.g. either use “and” before the last animal or remove it).

- Line 116: consider consolidating the figure 1 title e.g. map showing study location of NNP in relation to Kenya (left) and Africa (middle)…

- Line 126-129: consider re-writing sentence more concisely (e.g. Rodents were sampled using a line transect approach using a stratified random design…) or break-up into two sentences. The “for a given effort to sample rodents” component of the sentence doesn’t flow well.

- Line 129-130: this sentence is repetitive of the information provided in the sentence preceding it. Either remove this information from the first sentence which would make that sentence more concise, or remove this sentence.

- Line 130: replace “it is sensitive” with “they are sensitive”.

- Line 132: I think “a long” should be one word “along”.

- Line 133: insert “as” before “sites”. What is the distance from park boundaries that sampling occurred at?

- Line 136: what is the distance from edge boundaries that trapping occurred at?

- Line 140: sentence may benefit from adding “traps” after “relocating”.

- Line 143: a space is needed after the full stop. Consider replacing “of importance” with “recorded”. May also benefit from including that the species were recorded here too.

- Line 146: add s to “shrub”.

- Line 148, 150: consider using lower case m for the units as you have used lower case letters for other measurement units e.g. cm.

- Line 150: a space is needed after the full stop.

- Line 151: consider adding “possible” after “taxa”.

- Line 154, 156: keep formatting consistent when referring to “savannah, riverine, forest” – they appear as both capitals and lower case.

- Line 158, 248: the reference to figure 1 isn’t consistent (e.g. one is capitalised and one isn’t).

- Lines 164-167: the sentence doesn’t read properly with the inclusion of brackets around site names.

- Line 167: consider re-wording “site named” to “sites” and then replace the “was” (line 168) to “were”.

- Line 168: remove “savanna, forest and riverine” as is repetitive information.

- Line 174: remove the extra “ .”

- Line 180, 186, 191: is there relevant citations can include here?

- Line 217-222: sentence is too long and needs to either be broken up or re-written. Remove the repeated “the rodent species richness, abundance, and diversity” section.

- Line 231, 427, 430, 458: citation referenced incorrectly.

- Line 232: consider writing “ the smallest AIC value”.

- Line 235: consider writing” because non-lethal traps were used for…”

- Line 244: use the abbreviation for NNP.

- Line 245: add a comma after “J”.

- Line 246: add a full stop at the end of the sentence.

- Line 249: consider adding “sites” to the end of the sentence.

- Line 254: consider writing “where only a single species was caught” for improved clarity.

- Line 255: need to add in the close bracket.

- Lines 292, 293: add a “0” in front of the decimal point.

- Line 294: remove the comma after 6.

- Line 298, 312: should “per cent” read “percent”?

- Line 300: should it read “control sites”?

- Line 305: the results may benefit from adding in if the difference was significant or not.

- Line 306: add in “and” before “tree density”. Remove the capital from “Vegetation”.

- Line 308: it may be beneficial to define what you mean by dominant (e.g. was it based on a threshold of X% cover of a site?). This may be beneficial in the Methods.

- Line 309, 310: inconsistent formatting of savannah – sometimes with a capital, other times without.

- Line 311: was there any statistical analyses conducted for these comparisons? Was there significant differences in dominant plant species composition across the vegetation types? Report these.

- Line 326: remove “a” before “savannah”.

- Line 334: add “a” before “dominant tree”.

- Line 339-340: inconsistent formatting of sp (e.g. sp, sp and sp.).

- Line 348: add “more” before “abundant”.

- Line 349, 358, 368: suggest just referring to the supplementary table as “S1 Table” or “S2 Table” – relevant for all references to supplementary tables.

- Table 5: should the “wet season cf. dry season” be all on one line?

- Tables 6, 7, 8, 9: just wondering why the metrics are in all capitals (and table 9 also bold).

- Line 370: space needed in “Table7”.

- Table 9: should the forest for the second and third lines all be on one line?

- Line 392: “in” should have a capital I.

- Line 394: end bracket should not be in italics.

- Line 398: do the authors mean globally in reference to abundance or just within the study area?

- Line 404: remove “a” before “significant”.

- Line 405: this sentence is a very strong claim. I would suggest adding “can affect small mammal populations” as otherwise the sentence reads that this is always the case.

- Line 412: is there a citation for the distribution?

- Line 415: space required in “H.kerbispeterhansi”.

- Line 418: is there any evidence to support this claim in this study?

- Line 424: clarify whether this refers to in this study or in the aforementioned studies.

- Line 429: clarify whether this refers to native or introduced species or both.

- Line 453: perhaps a different word than “advances” would be better here.

- Line 456: perhaps add analyses or similar to the end of the sentence. Also, remove the “s” from “types”.

- Line 457: could you provide examples of the different resources required?

- Line 458: authors could expand on what they mean by “influences” e.g. positive/negative etc.

- Line 460: suggest re-writing this sentence to improve readability.

- Line 461: remove the “(“ in front of the square bracket.

- Line 465: perhaps add “species” after Mus.

- Line 471: the concluding paragraph may benefit from re-stating how the vegetation types influenced rodent diversity.

- Line 473: including the effect on abundance (e.g. increase/decrease) will help the reader finish the article with a clear understanding of how your measured metrics affected it. It may also be stronger take-home message to re-write the end of the sentence to something like “anthropogenic habitat disturbance has no direct influence”.

- References: spaces are required between the year, volume number and page numbers for the journal articles. Journal names also aren’t abbreviated as per PLOS ONE guidelines.

- Lines 534, 594, 601: have random numbers associated with them and no further information.

- Lines 628-630: suggest not including “.pdf” in the table title names.

- Figure titles: keep consistent in formatting throughout the manuscript – some ended in full stops while others didn’t.

Reviewer #5: The manuscript has been well written and all issues addressed, but I just have some few things to be addressed to improve the quality of the manuscript.

7. PLOS authors have the option to publish the peer review history of their article (what does this mean? ). If published, this will include your full peer review and any attached files.

**Do you want your identity to be public for this peer review?** For information about this choice, including consent withdrawal, please see our Privacy Policy .

Reviewer #4: No

Reviewer #5: No

---

## [Author Response · Author response to Decision Letter 3]

27 Sep 2024

Response to reviewer comments

We acknowledge the positive comments from Reviewers #1, 2, 3 and 5. We believe the valuable suggestions raised and addressed have contributed towards the improvement of the manuscript.

Reviewer #4: This manuscript provides novel research into the significance of ongoing monitoring of ecosystems which are disturbed by human development and activities, with results applicable to land managers. While the manuscript generally reads well, it could improve through the addition of some fine-scale detail and further explanation of the results and their implications. Furthermore, incorporating the key findings in relation to effects on rodent diversity and abundance into the concluding remarks will ensure that readers grasp the main findings of the study clearly. There are also formatting inconsistencies through the manuscript which require correction to enhance the readability of the paper. The referencing, both in text and in the reference list, still needs some attention to adhere to the PLOS ONE referencing guidelines.

We have now addressed the formatting inconsistencies throughout the manuscript in red, highlighting and reformatting all references according to the PLOS ONE journal guidelines. See track changes. In addition, we have clarified the key findings and implications in the concluding remarks.

Detailed comments can be found below.

- Appears there may be extra spaces in the text between words/after full stops e.g. Lines 35, 70, 99, 105, 111, 173, 203, 210, 251, 335 (space before the comma), 404.

Thank you for the correction. The extra spaces between words and full stops have been removed.

- Line 36: may read better as “were identified:”

Thank you. Noted and revised. See track changes.

- Lines 41, 46: instead of including that human disturbance had no effect on the two separate metrics, could have a sentence after stating what did have an effect on the metrics stating that human disturbance had no effect in both models.

Thank you, we have noted and revised the statement. See track changes.

- Line 48: add “measured” before “biotic and abiotic factors” to limit observations to those factors that were investigated in this study.

Thank you. Noted and revised. See track changes.

- Line 49: “a threat to the faunal diversity” – can the authors truly extrapolate their findings to all faunal diversity in the NNP with the same relationships they found for rodents occurring for all species?

Thank you for your valuable suggestion. We have looked into your comment and omitted this statement as it does not extrapolate to all faunal diversity for all species of rodents.

- Line 54: change the full stop after the citation to a comma.

Thank you. Noted and revised. See track changes.

- Line 59: what frequency of monitoring is required? Ongoing?

Thank you. We have added the frequency of monitoring which is ongoing (see track changes)

- Line 62: outline what “the choice” is referring to e.g. the choice of what?

Thank you. We refer to suitable bioindicator species.

- Line 64: remove the “a” before survival.

Thank you. Noted and revised. See track changes.

- Lines 66, 68, 84, 91: inconsistent formatting of “e.g.” (e.g e.g., and e,g.,) – ensure all are formatted the same throughout the text.

Thank you, we have now re/formatted abbreviation “e.g.” for consistency.

- Line 68: remove the “;” after “prey”.

Thank you, we have omitted “;” after “prey”.

- Lines 87, 89, 107: the citations have circular brackets instead of square brackets.

Thank you, all citations have been changed to square brackets.

- Line 92: is it better to say “park boundaries”?

Thank you. Noted and revised. See track changes.

- Line 98: start the sentence with “The”.

Thank you. We have started the sentence with “The”. See track changes.

- Line 104: add “a” before “decline”.

Thank you. Noted and added. See track changes.

- Line 110: “grass species, and scattered low-canopy” text shouldn’t be in italics.

Thank you, we have removed the italics. See track changes.

- Line113-114 and 224-225: consider keeping format the same in relation to the use of “and” in the list of animals within the two lots of brackets (e.g. either use “and” before the last animal or remove it).

Thank you we removed “and” as suggested. See track changes.

- Line 116: consider consolidating the figure 1 title e.g. map showing study location of NNP in relation to Kenya (left) and Africa (middle)…

Thank you, we have noted and revised.

- Line 126-129: consider re-writing sentence more concisely (e.g. Rodents were sampled using a line transect approach using a stratified random design…) or break-up into two sentences. The “for a given effort to sample rodents” component of the sentence doesn’t flow well.

Thank you, we have opted to break/down the sentence into two. See track changes.

- Line 129-130: this sentence is repetitive of the information provided in the sentence preceding it. Either remove this information from the first sentence which would make that sentence more concise, or remove this sentence.

Thank you, we have removed the sentence.

- Line 130: replace “it is sensitive” with “they are sensitive”.

Thank you, we noted that this statement had already been previously revised as suggested by the reviewer.

- Line 132: I think “a long” should be one word “along”.

Thank you, we have revised” a long” as one word.

- Line 133: insert “as” before “sites”. What is the distance from park boundaries that sampling occurred at?

Thank you, we have inserted “as” before “sites”. The distance from park boundaries was 100m.

- Line 136: what is the distance from edge boundaries that trapping occurred at?

The distance from edge boundaries that trapping occurred at was 50m.

- Line 140: sentence may benefit from adding “traps” after “relocating”.

Thank you for the comments we have now added “traps” after “relocating”. See track changes.

- Line 143: a space is needed after the full stop. Consider replacing “of importance” with “recorded”. May also benefit from including that the species were recorded here too.

Thank you for the suggestions. We have included space after full stop and replaced replacing “of importance” with “recorded” as well as included the recorded species.

- Line 146: add s to “shrub”.

Thank you for the comments we have now added s to “shrub”. See track changes.

- Line 148, 150: consider using lower case m for the units as you have used lower case letters for other measurement units e.g. cm.

Thank you, we have revised the format to included the right case for the abbreviation e.g. cm.

- Line 150: a space is needed after the full stop.

Thank you, we have added space after the full stop.

- Line 151: consider adding “possible” after “taxa”.

Thank you, we have added “possible” after “taxa”.

- Line 154, 156: keep formatting consistent when referring to “savannah, riverine, forest” – they appear as both capitals and lower case.

Thank you for the comment, we have now ensured that the formatting for “savannah, riverine, forest” –is consistent throughout the document. We used lower case. See track changes.

- Line 158, 248: the reference to figure 1 isn’t consistent (e.g. one is capitalised and one isn’t).

Thank you for the comment, we have edited the case as per you’re the reviewer’s suggestion. See track changes.

- Lines 164-167: the sentence doesn’t read properly with the inclusion of brackets around site names.

Thank you for the comment. We have opted to omit the content in brackets and correct the grammar for clarity,

- Line 167: consider re-wording “site named” to “sites” and then replace the “was” (line 168) to “were”.

Thank you for the comment, we have reworded “site named” to “sites” and replaced the “was” (line 168) to “were”.

- Line 168: remove “savanna, forest and riverine” as is repetitive information.

Thank you, we have removed the repeated information. See track changes.

- Line 174: remove the extra “ .”

Thank you, we have removed extra.

- Line 180, 186, 191: is there relevant citations can include here?

Thank you for your suggestion. We have looked into the reviewer comment and noted that we already included relevant citations.

- Line 217-222: sentence is too long and needs to either be broken up or re-written. Remove the repeated “the rodent species richness, abundance, and diversity” section.

Thank you for the comments. We have opted to broken down the sentence and removed redundant words.

- Line 231, 427, 430, 458: citation referenced incorrectly.

Thank you for the comment. We have now cited the references according to the PLOS ONE Journal guidelines.

- Line 232: consider writing “ the smallest AIC value”.

Thank you for the comment. We have written the smallest AIC value.

- Line 235: consider writing” because non-lethal traps were used for…”

Thank you for the comment. We have revised the statement as per the reviewer’s suggestion see track changes.

- Line 244: use the abbreviation for NNP.

Thank you for the comment. We have now used the abbreviation for NNP.

- Line 245: add a comma after “J”.

Thank you for the comment. We have added a comma after “J”.

- Line 246: add a full stop at the end of the sentence.

Thank you for the comment. a full stop at the end of the sentence.

- Line 249: consider adding “sites” to the end of the sentence.

Thank you for the comment. We have added “sites” to the end of the sentence.

- Line 254: consider writing “where only a single species was caught” for improved clarity.

Thank you for the comment. We have clarified the statement. See track changes.

- Line 255: need to add in the close bracket.

Thank you for the comment. We have added in the close bracket.

- Lines 292, 293: add a “0” in front of the decimal point.

Thank you for the comment. We have added a “0” in front of the decimal point.

- Line 294: remove the comma after 6.

Thank you for the comment. We have removed the comma after 6.

- Line 298, 312: should “per cent” read “percent”?

Thank you for the comment. We have written “per cent” as one word.

- Line 300: should it read “control sites”?

Thank you for the comment. Yes, it should read control sites.

- Line 305: the results may benefit from adding in if the difference was significant or not.

Thank you, we have noted that the level of significance is indicated clearly in Table 6.

- Line 306: add in “and” before “tree density”. Remove the capital from “Vegetation”.

Thank you, we have added in “and” before “tree density” and removed the capital from “Vegetation”.

- Line 308: it may be beneficial to define what you mean by dominant (e.g. was it based on a threshold of X% cover of a site?). This may be beneficial in the Methods.

Thank you for the comment. We have noted and revised in the methods section.

- Line 309, 310: inconsistent formatting of savannah – sometimes with a capital, other times without.

Thank you for the comment. We have now reformatted the savannah throughout the entire document. We decided to use lower case.

- Line 311: was there any statistical analyses conducted for these comparisons? Was there significant differences in dominant plant species composition across the vegetation types?

Thank you for the comment. All the statistics are included clearly in the results section and in the relevant Tables and figures.

Report these.

- Line 326: remove “a” before “savannah”.

Thank you, we have removed “a” before “savannah”.

- Line 334: add “a” before “dominant tree”.

Thank you, we have added “a” before “dominant tree”.

- Line 339-340: inconsistent formatting of sp (e.g. sp, sp and sp.).

We have now reformatted the sp throughout the entire document.

- Line 348: add “more” before “abundant”

Thank you, we have added “more” before “abundant”.

- Line 349, 358, 368: suggest just referring to the supplementary table as “S1 Table” or “S2 Table” – relevant for all references to supplementary tables.

Thank you, noted and revised.

- Table 5: should the “wet season cf. dry season” be all on one line?

Thank you, noted and revised.

- Tables 6, 7, 8, 9: just wondering why the metrics are in all capitals (and table 9 also bold).

Thank you for the comment. We have revised the tables 6, 7, 8 and 9 to have the same format.

- Line 370: space needed in “Table7”.

We have included space in “Table7”.

- Table 9: should the forest for the second and third lines all be on one line?

Thank you, noted and revised.

- Line 392: “in” should have a capital I.

We have capitalized I.

- Line 394: end bracket should not be in italics.

This was an oversight on our part, we have removed italics on the end bracket.

- Line 398: do the authors mean globally in reference to abundance or just within the study area?

Thank you for the comment. The authors refer to the tropics which we have added in the manuscript.

- Line 404: remove “a” before “significant”.

Thank you, we have removed “a” before “significant”.

- Line 405: this sentence is a very strong claim. I would suggest adding “can affect small mammal populations” as otherwise the sentence reads that this is always the case.

Thank you for the comment. We have noted and revised the statement as per your suggestion.

- Line 412: is there a citation for the distribution?

Noted, we have included the citation for the distribution.

- Line 415: space required in “H.kerbispeterhansi”.

Thank you we have added space in “H.kerbispeterhansi” as it is supposed to be written in scientific nomenclature. We have also italicized it.

- Line 418: is there any evidence to support this claim in this study?

Thank you for the comment. We have previously provided the evidence to support the study.

- Line 424: clarify whether this refers to in this study or in the aforementioned studies.

Thank you for the comment. We have clarified that this refers to the aforementioned studies. See track changes.

- Line 429: clarify whether this refers to native or introduced species or both.

Thank you for the comment. This refers to both native and introduced species.

- Line 453: perhaps a different word than “advances” would be better here.

Thank you, we have noted and revised. We have opted to use “promotes” instead of “advances”.

- Line 456: perhaps add analyses or similar to the end of the sentence. Also, remove the “s” from “types”.

- Line 457: could you provide examples of the different resources required?

Thank you for your comment, we have provided the different resources as required by the reviewer. See track changes.

- Line 458: authors could expand on what they mean by “influences” e.g. positive/negative etc.

We have opted not to use “influences” e.g. positive/negative etc but instead rephrase and use impacting considering high forest cover was linked to high rodent diversity with regards to species and vice-versa.

- Line 460: suggest re-writing this sentence to improve readability.

Thank you for your comment. We have re-written the sentence. See track changes.

- Line 461: remove the “(“ in front of the square bracket.

Thank you, we have now removed the “(“in front of the square bracket. See track changes.

- Line 465: perhaps add “species” after Mus.

Thank you, we have now added “species” after Mus. See track changes.

- Line 471: the concluding paragraph may benefit from re-stating how the vegetation types influenced rodent diversity.

Thank you for the valuable suggestion, we have now added the information on how vegetation types influenced rodent diversity on our study.

- Line 473: including the effect on abundance (e.g. increase/decrease) will help the reader finish the article with a clear understanding of how your measured metrics affected it. It may also be stronger take-home message to re-write the end of the sentence to something like “anthropogenic habitat disturbance has no direct influence”.

Thank you for your valuable suggestion. We agree with the reviewer’s comments and have re-written the take-home message.

- References: spaces are required between the year, volume number and page numbers for the journal articles.

---

## [Decision Letter · Decision Letter 3]

21 Jan 2025

PONE-D-23-23260R3Drivers of Rodent Community Structure in an Urban National Park, KenyaPLOS ONE

Dear Dr. Lwande,

Thank you for submitting your manuscript to PLOS ONE. After careful consideration, we feel that it has merit but does not fully meet PLOS ONE’s publication criteria as it currently stands. Therefore, we invite you to submit a revised version of the manuscript that addresses the points raised during the review process.

We look forward to receiving your revised manuscript.

Kind regards,

Clement Ameh Yaro, Ph.D

Academic Editor

PLOS ONE

Journal Requirements:

Reviewers' comments:

Reviewer's Responses to Questions

**Comments to the Author**

1. If the authors have adequately addressed your comments raised in a previous round of review and you feel that this manuscript is now acceptable for publication, you may indicate that here to bypass the “Comments to the Author” section, enter your conflict of interest statement in the “Confidential to Editor” section, and submit your "Accept" recommendation.

Reviewer #5: All comments have been addressed

Reviewer #6: All comments have been addressed

2. Is the manuscript technically sound, and do the data support the conclusions?

Reviewer #5: Yes

Reviewer #6: Yes

3. Has the statistical analysis been performed appropriately and rigorously? 

Reviewer #5: Yes

Reviewer #6: Yes

4. Have the authors made all data underlying the findings in their manuscript fully available?

Reviewer #5: Yes

Reviewer #6: Yes

5. Is the manuscript presented in an intelligible fashion and written in standard English?

Reviewer #5: Yes

Reviewer #6: Yes

6. Review Comments to the Author

Reviewer #5: All authors have answered queries raised with regards to improving this manuscript. I want to say to the authors, Well done!

Reviewer #6: The manuscript focuses on the rodent assemblage in a national park embedded within an urban matrix, making it highly relevant for understanding the wildlife in these areas, which are essential for biodiversity conservation. The manuscript is well-written and, from what I can see, has improved substantially since the initial version submitted. I only have two minor comments:

- I could not find information about the years and months when the rodent sampling was conducted. I suggest adding this information.

- It would be helpful to include details about the size of the national park (in km²) for readers who may not be familiar with it.

7. PLOS authors have the option to publish the peer review history of their article (what does this mean? ). If published, this will include your full peer review and any attached files.

**Do you want your identity to be public for this peer review?** For information about this choice, including consent withdrawal, please see our Privacy Policy .

Reviewer #5: No

Reviewer #6: No

---

## [Author Response · Author response to Decision Letter 4]

27 Jan 2025

Clement Ameh Yaro, Ph.D

Academic Editor

PLOS ONE

27th January 2025

Dear Clement Ameh Yaro, Ph.D,

RE: Response to reviewer comments on Manuscript Number: PONE-D-23-23260R3-Drivers of Rodent Community Structure in an Urban National Park, Kenya

We appreciate your time and work in ensuring we receive relevant feedback from all reviewers designated #5 and #6 on the aforementioned manuscript. The comments have helped us improve the manuscript. Please find a point-by-point response to each of the reviewers' comments and suggestions in red text below. We have also altered it with track adjustments. We hope the amended version meets the reviewers' and editorial team's expectations.

Thank you for your unwavering support.

Sincerely,

On behalf of all authors,

Olivia Wesula Lwande, corresponding author,

Department of Clinical Microbiology, Virology, 901-85

SE, University of Umeå, Sweden

olivia.lwande@umu.se

Journal Requirements:

Thank you for your comment. We have reviewed all the cited references and the reference list to ensure it conforms to your esteemed journal PLOS ONE reference guidelines.

Reviewers' comments:

Reviewer's Responses to Questions

Comments to the Author

1. If the authors have adequately addressed your comments raised in a previous round of review and you feel that this manuscript is now acceptable for publication, you may indicate that here to bypass the “Comments to the Author” section, enter your conflict of interest statement in the “Confidential to Editor” section, and submit your "Accept" recommendation.

Reviewer #5: All comments have been addressed

Thank you for taking the time to review our manuscript. We are pleased to know that we have adequately addressed all the comments as per the reviewers' suggestions.

Reviewer #6: All comments have been addressed

We thank the reviewer for the time and effort invested in reviewing our manuscript. We are glad to know that we have managed to work on the comments raised satisfactorily.

2. Is the manuscript technically sound, and do the data support the conclusions?

Reviewer #5: Yes

Reviewer #6: Yes

We appreciate the positive comments from reviewers #5 and #6 concerning our manuscript being technically sound and appropriately drawing our conclusion based on the data demonstrated.

3. Has the statistical analysis been performed appropriately and rigorously?

Reviewer #5: Yes

Reviewer #6: Yes

Thank you for your positive response to the questions. We acknowledge that reviewers #5 and #6 have approved that the statistical analysis has been performed appropriately and rigorously.

4. Have the authors made all data underlying the findings in their manuscript fully available?

Reviewer #5: Yes

Reviewer #6: Yes

Thank you for the reply. We have provided all the data representing all the study findings. This has also been confirmed by reviewers #5 and #6.

5. Is the manuscript presented in an intelligible fashion and written in standard English?

Reviewer #5: Yes

Reviewer #6: Yes

Thank you for your positive reply. We agree with reviewers #5 and # 6 that the language and grammar used in the manuscript is in standard English. All the typographical and grammatical errors were adequately addressed during the later review process (version R3).

6. Review Comments to the Author

Reviewer #5: All authors have answered queries raised with regards to improving this manuscript. I want to say to the authors, Well done!

Thank you very much for the positive comment and for supporting our manuscript. We appreciate your dedication and scientific input in improving the manuscript.

Reviewer #6: The manuscript focuses on the rodent assemblage in a national park embedded within an urban matrix, making it highly relevant for understanding the wildlife in these areas, which are essential for biodiversity conservation. The manuscript is well-written and, from what I can see, has improved substantially since the initial version submitted.

Thank you for the encouraging comment. We acknowledge that indeed the manuscript has improved a lot due to your valuable input. We appreciate all your effort and hope that it will meet the expectations of the editorial team.

I only have two minor comments:

- I could not find information about the years and months when the rodent sampling was conducted. I suggest adding this information.

Thank you for the comment. Rodent sampling was conducted from December 2020 to June 2021 (see page 7 line 126) in the materials and methods section under the sub-heading entitled “Rodent community survey”.

- It would be helpful to include details about the size of the national park (in km²) for readers who may not be familiar with it.

The Nairobi National Park is 117 km² (see page 6 line 99 in the materials and methods under subheading entitled “study area”. This is supported by reference

“Owino AO, Kenana ML, Webala P, Andanje S, Omondi PO. Patterns of variation of herbivore assemblages at Nairobi National Park, Kenya, 1990-2008. Journal of Environmental Protection. 2011;02(1):855-866.”

7. PLOS authors have the option to publish the peer review history of their article (what does this mean?). If published, this will include your full peer review and any attached files.

All authors agree to the option to publish the peer-review history of our article for transparency and accountability.

Do you want your identity to be public for this peer review? For information about this choice, including consent withdrawal, please see our Privacy Policy.

Reviewer #5: No

Reviewer #6: No

We respect the reviewers' decision to remain anonymous.

---

## [Decision Letter · Decision Letter 4]

10 Mar 2025

Drivers of Rodent Community Structure in an Urban National Park, Kenya

PONE-D-23-23260R4

Dear Dr. Lwande,

We’re pleased to inform you that your manuscript has been judged scientifically suitable for publication and will be formally accepted for publication once it meets all outstanding technical requirements.

Kind regards,

Clement Ameh Yaro, Ph.D

Academic Editor

PLOS ONE

Additional Editor Comments (optional):

Reviewers' comments:

Reviewer's Responses to Questions

**Comments to the Author**

1. If the authors have adequately addressed your comments raised in a previous round of review and you feel that this manuscript is now acceptable for publication, you may indicate that here to bypass the “Comments to the Author” section, enter your conflict of interest statement in the “Confidential to Editor” section, and submit your "Accept" recommendation.

Reviewer #5: All comments have been addressed

Reviewer #6: All comments have been addressed

2. Is the manuscript technically sound, and do the data support the conclusions?

Reviewer #5: Yes

Reviewer #6: Yes

3. Has the statistical analysis been performed appropriately and rigorously? 

Reviewer #5: Yes

Reviewer #6: Yes

4. Have the authors made all data underlying the findings in their manuscript fully available?

Reviewer #5: Yes

Reviewer #6: Yes

5. Is the manuscript presented in an intelligible fashion and written in standard English?

Reviewer #5: Yes

Reviewer #6: Yes

6. Review Comments to the Author

Reviewer #5: (No Response)

Reviewer #6: Thank you for addressing all the comments. Congratulations for your work, very interesting for the conservation of biodiversity in natural spaces within urban environments.

7. PLOS authors have the option to publish the peer review history of their article (what does this mean? ). If published, this will include your full peer review and any attached files.

**Do you want your identity to be public for this peer review?** For information about this choice, including consent withdrawal, please see our Privacy Policy .

Reviewer #5: **Yes: ** Olukayode James Adelaja

Reviewer #6: No

---

## [Editor Report · Acceptance letter]

PONE-D-23-23260R4

PLOS ONE

Dear Dr. Lwande,

I'm pleased to inform you that your manuscript has been deemed suitable for publication in PLOS ONE. Congratulations! Your manuscript is now being handed over to our production team.

Kind regards,

on behalf of

Dr. Clement Ameh Yaro

Academic Editor

PLOS ONE